# Improving Single-round Active Adaptation:
# A Prediction Variability Perspective

**Xiaoyang Wang** *  
*University of Illinois Urbana-Champaign*

*xw28@illinois.edu*

**Yibo Jacky Zhang**  
*Stanford University*

*yiboz@stanford.edu*

**Olawale Salaudeen**  
*Massachusetts Institute of Technology*

*olawale@mit.edu*

**Mingyuan Wu**  
*University of Illinois Urbana-Champaign*

*mw34@illinois.edu*

**Hongpeng Guo**  
*University of Illinois Urbana-Champaign*

*hg5@illinois.edu*

**Chaoyang He**  
*TensorOpera, Inc.*

*ch@tensoropera.com*

**Klara Nahrstedt**  
*University of Illinois Urbana-Champaign*

*klara@illinois.edu*

**Sanmi Koyejo**  
*Stanford University*

*sanmi@cs.stanford.edu*

**Reviewed on OpenReview:** `https://openreview.net/forum?id=Vthqn5VE7L`

## Abstract

Machine learning models trained with offline data often suffer from distribution shifts in online environments and require fast adaptation to online data. The high volume of online data further stimulates the study of active adaptation approaches that achieve competitive adaptation performance by selectively annotating only 5%-10% of online data and using it to continuously train a model. Despite the reduction in data annotation cost, many prior active adaptations assume a multi-round data annotation procedure during continuous training, which hinders timely adaptation. In this work, we study a single-round active adaptation problem with a minimum data annotation turnaround time but require the selected subset of data samples to help the entire continuous training procedure until convergence. In our theoretical analysis, we find that the prediction variability of each data sample throughout the

---

*Work partially performed while at TensorOpera (formerly FedML) and done before XW started at Amazon.

training is crucial, in addition to the conventional data diversity. The prediction variability measures how much the prediction could possibly change during the continuous training procedure. To this end, we introduce a novel approach called feature-norm scaled gradient embedding (FORGE), which incorporates prediction variability and improves the single-round active adaptation performance when combined with standard data selection strategies (e.g., k-center greedy). In addition, we provide efficient implementations to construct our FORGE embedding analytically without explicitly backpropagating gradients. Empirical results further demonstrate that our approach consistently outperforms the random selection baseline by up to 1.26% for various vision and language tasks while other competitors often underperform the random selection baseline.

# 1 Introduction

The data in production environments can shift away from what is used for training the model. For example, a vision model inside a camera of a surveillance or autonomous driving system may see new images that are different from its offline training images every day. A fraud detection model may process emails and transactions from new users or adversaries that try to penetrate novel attacks to bypass the detection. A language model in a chat application also receives new and time-based questions over time. Such ubiquitous distribution shifts are among the major causes of performance degradation in machine learning models (Huyen, 2022). The consequence of performance degradation caused by distribution shifts can be quite severe: a failure of a vision model may cause traffic accidents, penetrating a novel fraud can cause financial loss to a company, and a language model generating incorrect answers to new questions can raise concerns in mission-critical applications such as medical diagnostic and healthcare.

One of the most effective ways to address distribution shift problems is continuously training a model using online data (Huyen, 2022). Given a large amount of online data and the subsequent annotation cost, a few recent works (Prabhu et al., 2021; Xie et al., 2023) explore active adaptation and show that carefully curating a subset of online data is an effective means to achieve superior adaptation performance while significantly reducing the data annotation cost. Despite lowering the annotation cost by a factor of ten or twenty, existing active adaptation methods assume multi-round data annotation procedures. Under these multi-round settings, we issue multiple queries that sequentially request labels for selected data samples from data annotators and continuously train a model between adjacent queries for a few iterations. However, these sequentially issued queries may hinder timely adaptation, causing user dissatisfaction.

Language models interact with users and continuously improve their performance based on user feedback (e.g., preference annotations). Repeatedly asking users to annotate their preferences without delivering a customized experience can hurt their satisfaction. Besides, for a conventional supervised learning problem (e.g., fraud detection), issuing $K$ mini-batches of queries can incur $\sim K$ times more turnaround time than selecting data samples once and issuing a single batch of queries because modern data annotation systems (e.g., MTurk (Crowston, 2012)) are optimized for throughout (Haas et al., 2015; Difallah et al., 2015). The throughput-oriented systems are good at handling a large batch instead of multiple mini-batches. To this end, we propose a single-round active adaptation problem, where we select once and continuously train a model for many iterations until convergence (Figure 1).

The single-round adaptation problem requires us to study whether the selected subset is helpful throughout the continuous training procedure with many iterations. To this end, we start with a pair-wise loss reduction gap that measures how much learning one selected sample can help learn another unselected sample. We also show that reducing this gap can improve adaptation performance. Then, we show that the loss reduction gap depends on (1) gradient distance, which is important for the first few continuous training iterations, and (2) prediction variability, which can dominate the latter iterations. This prediction variability is estimated using the norm of a tangent feature of a linearized neural network, which is a plausible model for adaptation tasks with a few fine-tuning epochs (Malladi et al., 2023). The tangent feature characterizes how the prediction varies during continuous training, and the norm of the tangent feature can upper bound the variability.

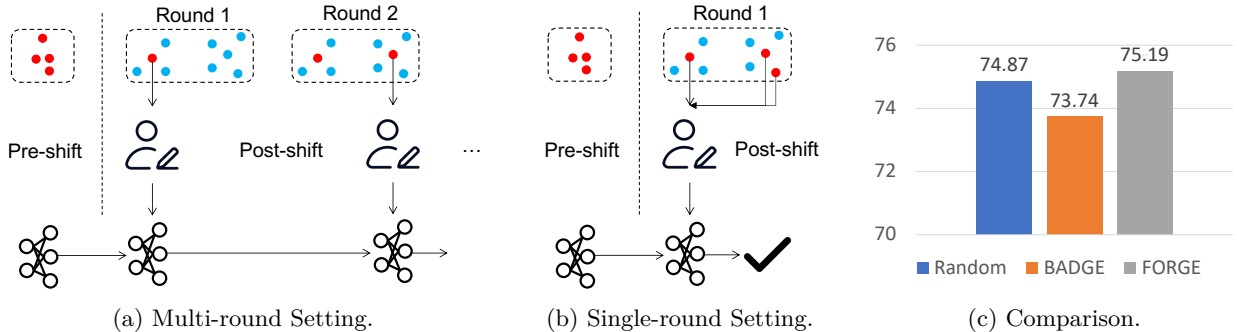

(a) Multi-round Setting.  (b) Single-round Setting.  (c) Comparison.

Figure 1: Prior works apply active learning to reduce the data annotation cost of adapting machine learning models to shifting distributions. However, many assume a multi-round setting (a), which incurs multiple turnaround times and unnecessary delays. We study a single-round adaptation setting (b) and develop an improved approach called feature-norm scaled gradient embedding (FORGE) (c).

Based on the theoretical analysis, we develop a feature-norm scaled gradient embedding (FORGE) – an improved active adaptation approach that considers the prediction variability and achieves better single-round active adaptation performance. One interesting observation from our analysis is that certain data samples may have small gradient distances and show a small loss reduction gap in the first few iterations. However, their loss reduction gap may increase significantly with high prediction variability. Following this observation, we aim to discover data samples with small gradient distances but high variability. Specifically, our approach first represents each data sample using their gradient embeddings, defined as the gradient of the loss w.r.t. the parameters and can help represent the gradient distance. We then re-scale the gradient embedding according to the sample-wise prediction variability, measured by the tangent feature norm, and construct feature-norm scaled gradient embedding. With the re-scaling operation, high-variability samples are represented by "long" embedding vectors. Therefore, they are more likely to be picked by diversity-based data acquisition methods such as k-center greedy that primarily look for long vectors far away from the selected ones (Ash et al., 2020). We outline suffucient conditions under which our feature-norm scaled embeddings outperform an approach without feature-norm scaling.

We derive an efficient analytical implementation for FORGE embedding that eliminates the need for expensive gradient backpropagation. We also extend the derivation to various vision and language tasks, including image classification, sentence classification, span-based question and answering, and reward modeling. Extensive empirical evaluation further demonstrates the advantage of our approach for single-round active adaptation tasks. Our main contribution is listed as follows:

- We comprehensively analyze a single-round adaptation problem, outlining two key conditions: gradient distance and prediction variability.

- We develop an improved active adaptation approach by incorporating prediction variability.

- We provide efficient implementations and extensively evaluate them with various tasks.

## 2 Related Work

**Active learning.**  Common active learning approaches are based on data diversity and data uncertainty. Diversity-based active learning (Sener & Savarese, 2018; Ash et al., 2020; Shen et al., 2022) aims to select a subset of diverse data samples that can best represent the full dataset in the input space, an embedding space, or a gradient embedding space. Uncertainty-based active learning (Balcan et al., 2007; Gal & Ghahramani, 2016; Ban et al., 2022; 2024) prioritizes the selection of data samples where the model prediction is uncertain. Such uncertainty often requires entropy estimation and careful calibration. A few recent works also show that incorporating training dynamics can improve convergence speed (Wang et al., 2022; Mohamadi et al., 2022). Most of these approaches interleave the data selection and the model training procedure and perform

selection every few training iterations. In contrast, our work studies single-round data selection with many training iterations for timely adaptation. Chen et al. (2022); Wang et al. (2023) studied similar single-round problems to ours but assumed learning from scratch instead of adaptation. Several recent works apply active learning to foundations model training (Zhang et al., 2023; Bhatt et al., 2024; Shen et al., 2025), and we will show a case with reward modeling as part of our experiments.

**Domain adaptation.** Li et al. (2021); Zhao et al. (2022) demonstrated the difficulty of unsupervised adaptation. Hence, recent works have focused on incorporating external supervision (Su et al., 2020; Prabhu et al., 2021; Li et al., 2021; Zhou et al., 2022; Xie et al., 2023; Alabdulmohsin et al., 2023; Tsai et al., 2024). However, most of the works on active adaptation focus on improving the performance of conventional active learning techniques via, for example, balancing diversity and uncertainty (Prabhu et al., 2021) or improving uncertainty calibration (Xie et al., 2023), but few aim to minimize the annotation turnaround time.

## 3 Preliminaries

We list the notation and introduce an active adaptation algorithm framework before proceeding with the technical discussion. Appendix A provides a table of notations to ease the reading further.

### 3.1 Notation

A pair of input $\boldsymbol{x} \in \mathcal{X}$ and its label $y \in \mathcal{Y}$ is a data sample. $f_\theta : \mathcal{X} \to \mathcal{Y}$ is a model (e.g., neural network) that is parameterized by $\theta$. $\ell : \mathcal{Y} \times \mathcal{Y} \to \mathbb{R}$ is a loss function. $\boldsymbol{z}$ is the last hidden state of a neural network (i.e., input to the last linear layer). We use $\ell(\boldsymbol{x}, y; f_\theta)$ to simplify the notation of $\ell(f_\theta(\boldsymbol{x}), y)$, which denotes the loss of a function $f_\theta$ at a given data sample $(\boldsymbol{x}, y)$. $r_{0 \to T}(\boldsymbol{x}, y; f_\theta) = \ell(\boldsymbol{x}, y; f_{\theta_0}) - \ell(\boldsymbol{x}, y; f_{\theta_T})$ is the amount of loss reduction on a data sample $(\boldsymbol{x}, y)$ after the model parameter evolves from $\theta_0$ to $\theta_T$ after $T$ training iterations. $\nabla_\theta f_\theta(\boldsymbol{x})$ is the gradient of the model output $f_\theta(\boldsymbol{x})$ w.r.t. the model parameter $\theta$, which is also called a tangent feature (Jacot et al., 2018; Lee et al., 2019) in the following sections. $\nabla_\theta \ell(\boldsymbol{x}, y; f_\theta)$ is the gradient of the loss value w.r.t. the model parameter $\theta$. $\| \cdot \|$ denotes the $L_2$ norm. $\boldsymbol{s}$ denotes a selected subset of indexes from the full set. $| \cdot |$ denotes the size of a set. $[n]$ denotes a set of $n$ natural numbers. $\mathcal{S}$ is a set of selected data samples $\{\boldsymbol{x}_i, y_i \mid i \in \boldsymbol{s}\}$, $\mathcal{S}'$ is a set of unselected data samples $\{\boldsymbol{x}_i, y_i \mid i \in [n] \setminus \boldsymbol{s}\}$ and $\hat{\mathcal{S}}$ is a set of unselected data samples and its closet selected neighbor $\{\boldsymbol{x}_i, y_i, \boldsymbol{x}_j, y_j \mid i \in [n] \setminus \boldsymbol{s}, j = \arg\min_{\boldsymbol{s}} \|\phi(\boldsymbol{x}_i) - \phi(\boldsymbol{x}_j)\|\}$. $\phi$ is an embedding function. $\mathrm{Cat}(\cdot, \cdot)$ is a vector concatenation operator.

### 3.2 Algorithm Framework

Our approach operates under a conventional two-step active learning procedure (Sener & Savarese, 2018; Ash et al., 2020): (1) construct data representations using an embedding function $\phi$ and (2) perform diversity-based data selection. We use the k-center greedy algorithm (lines 3-6) as an example in the algorithm framework (Algorithm 1). Then, we continuously train a model using the selected data samples. The k-center greedy algorithm can effectively minimize the maximum distance between an unselected data sample $\boldsymbol{x}'$ and its closest selected neighbor $\boldsymbol{x}$, which can be formulated as an objective function $\max_{i \in [N] \setminus \boldsymbol{s}} \min_{j \in \boldsymbol{s}} \|\boldsymbol{x}_i - \boldsymbol{x}_j\|$ (Sener & Savarese, 2018). Intuitively, this k-center greedy algorithm selects data samples that are far from others. Our framework differs from prior ones (Ash et al., 2020) regarding the continuous training step (line 7), which was often placed inside the data selection loop (lines 3-6).

## 4 Analysis

This section introduces our main insights into the single-round active adaptation problem. Our goal is to select a subset of data samples; once a model is trained upon them for many iterations, the model performance is comparable to that of a model trained over the full dataset. This goal requires us to study whether learning with selected data samples can also help learn the unselected ones, quantified by a loss reduction gap. We show that such a loss reduction gap is vital in learning the full dataset via a subset. However, estimating the loss reduction gap after many training iterations is non-trivial because the model's training dynamics remain

---

**Algorithm 1** Algorithm framework

---

**Input:** A set of data samples $\{\boldsymbol{x}_1, ..., \boldsymbol{x}_N\}$, an embedding function $\phi$, and a model $f_{\theta_0}$.
**Steps:**
1: Construct data representations $\{\phi(\boldsymbol{x}_1), ..., \phi(\boldsymbol{x}_N)\}$ using an embedding function $\phi$;
2: Initialize a set of selected indices $\boldsymbol{s} = \{s_1\}$ with a random $s_1 \sim \text{unif}(0, N)$;
3: **for** $k \leftarrow 2$ to $K$ **do**
4:     $s_k = \arg\max_{i \in [N] \setminus \boldsymbol{s}} \min_{j \in \boldsymbol{s}} \|\phi(\boldsymbol{x}_i) - \phi(\boldsymbol{x}_j)\|$;
5:     $\boldsymbol{s} \leftarrow \boldsymbol{s} \cup \{s_k\}$;
6: **end for**
7: Continuously training a model $f_{\theta_0}$ by minimizing $\frac{1}{\|\boldsymbol{s}\|} \sum_{i=1}^{k} \ell(\boldsymbol{x}_{\boldsymbol{s}_i}, y_{\boldsymbol{s}_i}; f_\theta)$ for $T$ iterations;
8: Return $f_{\theta_T}$.

---

unknown at the selection step. To this end, we introduce a novel method to estimate an upper bound of the loss reduction gap incorporating (1) a gradient distance term and (2) a prediction variability term under unknown training dynamics.

By obtaining a loss reduction gap upper bound, we observe that minimizing the upper bound has a sample-wise difficulty, depending not only on the gradient distance between samples but also on a sample-wise prediction variability. Moreover, the impact of the prediction variability can grow quadratically as the model parameter deviates from its initialization during training. The quadratic growth of the prediction variability can dominate the term associated with the gradient distance, which only grows linearly. Such a result goes beyond the prior result (Sener & Savarese, 2018) on the distance term and will further guide our algorithm design in Section 5.

Technically, we employ the linearization of a non-linear neural network with a mean squared error loss function under a neural tangent kernel (NTK) regime (Jacot et al., 2018; Lee et al., 2019; Malladi et al., 2023); a standard tool in deep learning theoretical analysis (Ren & Sutherland, 2025). Notably, this approach does not require assuming linear models. The mean squared error loss function is the standard choice in the NTK regime (Jacot et al., 2018; Lee et al., 2019; Malladi et al., 2023), which produces clear theoretical results and behaves closely to the cross-entropy loss function (Hui & Belkin, 2021). This NTK regime is increasingly employed in modern active learning research (Awasthi et al., 2021; Mohamadi et al., 2022; Wang et al., 2022).

### 4.1 Objective Function

In an active adaptation problem, we hope that learning one selected data sample $\boldsymbol{x}$ can also help learning another unselected data sample $\boldsymbol{x}'$. To this end, we introduce an objective function called loss reduction gap between a pair of selected and unselected data samples.

**Definition 1.** *(Loss reduction gap) Let $r_{0 \to T}(\boldsymbol{x}, y; f_\theta) = \ell(\boldsymbol{x}, y; f_{\theta_0}) - \ell(\boldsymbol{x}, y; f_{\theta_T})$ be the loss reduction on a data sample $(\boldsymbol{x}, y)$ after the model parameter evolves from $\theta_0$ to $\theta_T$ after $T$ training iterations , we define a loss reduction gap between $\boldsymbol{x}$ and $\boldsymbol{x}'$:*

$$r_{0 \to T}(\boldsymbol{x}, y; f_\theta) - r_{0 \to T}(\boldsymbol{x}', y'; f_\theta). \tag{1}$$

The loss reduction gap includes the loss reduction $r_{0 \to T}(\boldsymbol{x}, y; f_\theta)$ on a selected data sample $x$ and the loss reduction $r_{0 \to T}(\boldsymbol{x}', y'; f_\theta)$ on an unselected data sample $x'$. In our active adaptation problem, for a given amount of loss reduction on a selected $x$, we hope that the model $f_{\theta_T}$ parameterized by $\theta_T$ also learns $x'$ and reduces $\ell(\boldsymbol{x}', y'; f_{\theta_T})$. Note that the loss reduction is an objective, and we do not assume that the reduction is positive. The following proposition further illustrates the role of the loss reduction gap in minimizing the expected loss reduction $\mathbb{E}_\mathcal{D}[r_{0 \to T}(\boldsymbol{x}, y; f)]$ with a given data distribution $\mathcal{D}$.

**Proposition 2.** *(Decomposition of expected loss reduction) Let $\boldsymbol{x}$ be the closest selected neighbor of an unselected $\boldsymbol{x}'$ and $w_i = c_i + 1$ where $c_i$ is the frequency of each $\boldsymbol{x}$ appears as the closest neighbor, the expected*

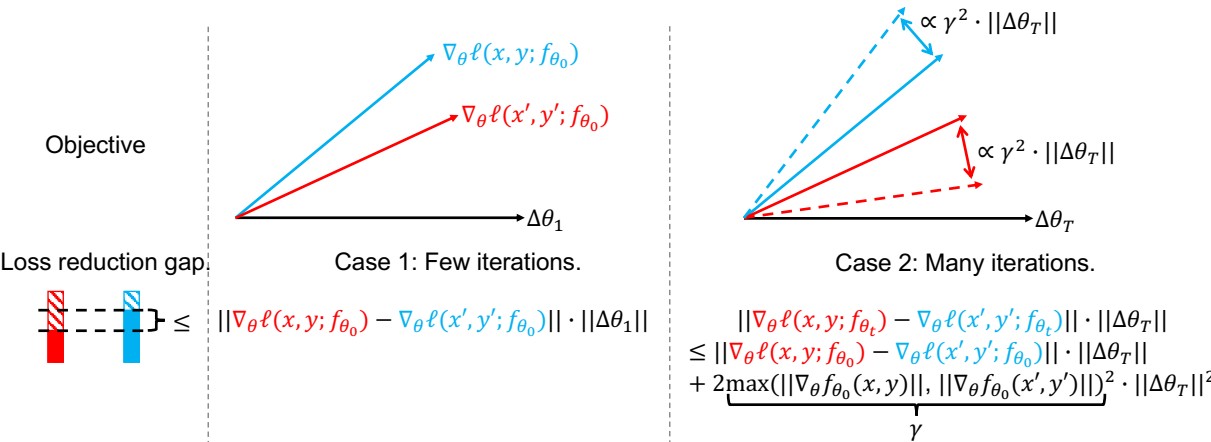

Figure 2: The gradient similarity $\nabla_\theta \ell(\boldsymbol{x}, y; f_{\theta_0}) - \nabla_\theta \ell(\boldsymbol{x}', y'; f_{\theta_0})$ can indicate a loss reduction gap in a few training iterations setting (case 1). With an increasing amount of training iterations (case 2), we must also consider the prediction variability (i.e., $\gamma = \max(\|\nabla_\theta f_{\theta_0}(\boldsymbol{x})\|, \|\nabla_\theta f_{\theta_0}(\boldsymbol{x}')\|)$).

*loss reduction $\mathbb{E}_{\mathcal{D}}[r_{0 \to T}(\boldsymbol{x}, y; f_\theta)]$ with a given data distribution $\mathcal{D}$ can be decomposed and upper bounded by (1) a training loss reduction, (2) a maximum loss reduction gap, and (3) a generalization gap:*

$$
\begin{aligned}
&\mathbb{E}_{\mathcal{D}}[r_{0 \to T}(\boldsymbol{x}, y; f_\theta)] \\
&\geq \underbrace{\frac{1}{N} \sum_{i \in \boldsymbol{s}} w_i r_{0 \to T}(\boldsymbol{x}_i, y_i; f_\theta)}_{\textit{Weighted training loss reduction}} + \underbrace{\mathbb{E}_{\mathcal{D}}[r_{0 \to T}(\boldsymbol{x}, y; f_\theta)] - \frac{1}{N} \sum_{i=1}^{N} r_{0 \to T}(\boldsymbol{x}_i, y_i; f)}_{\textit{Generalization gap}} \\
&\quad - \frac{1}{N} \cdot \underbrace{\max_{\boldsymbol{x}', y', \boldsymbol{x}, y \in \hat{\mathcal{S}}} |r_{0 \to T}(\boldsymbol{x}, y; f_\theta) - r_{0 \to T}(\boldsymbol{x}', y'; f_\theta)|}_{\textit{Maximum loss reduction gap}}.
\end{aligned}
\tag{2}
$$

The expected loss reduction $\mathbb{E}_{\mathcal{D}}[r_{0 \to T}(\boldsymbol{x}, y; f_\theta)]$ on the left-hand-side of Equation 2 is what we want to maximize but lacks direct estimation of it. Thanks to the lower bound in Proposition 2, we obtain additional insights suggesting that the expected loss reduction is lower bounded by (1) how much does the loss on the selected samples reduce, (2) the standard generalization gap between the data distribution and the data samples and (3) the loss reduction gap between the selected and unselected data samples in an active learning procedure. In what follows, we will present a new upper bound of the loss reduction gap. The upper bound is helpful because reducing the upper bound of the loss reduction gap can help increase the lower bound of the expected loss reduction $\mathbb{E}_{\mathcal{D}}[r_{0 \to T}(\boldsymbol{x}, y; f_\theta)]$.

## 4.2 Main Result

Estimating the loss reduction gap is non-trivial because the model parameter $\theta_T$ at iteration $T$ is unknown at the data selection step. Therefore, we derive an upper bound of the loss reduction gap applicable to unknown $\theta_T$. Our result suggests that for an arbitrary $\theta_T$, the upper bound depends on the gradient similarity between a pair of data samples and their prediction variability. The gradient similarity term complements prior practice (Ash et al., 2020), and the prediction variability term will guide an improved algorithm design. Before proceeding to the upper bound, we first introduce the NTK regime that is used in our analysis.

**Assumption 3.** *(NTK regime) (Jacot et al., 2018; Lee et al., 2019) Let $f_{\theta_T} : \mathcal{X} \to \mathcal{Y}$ be a non-linear model at training step $T$, we assume that its output $f_{\theta_T}(\boldsymbol{x})$ are governed by a linear model $f^{\text{lin}} : \mathcal{X} \to \mathcal{Y}$ obtained*

*from the first-order Taylor expansion of the non-linear model $f_{\theta_T}$ around its initial parameter $\theta_0$:*

$$f_{\theta_T}(\boldsymbol{x}) \approx f_{\theta_T}^{\mathrm{lin}}(\boldsymbol{x}) = f_{\theta_0}(\boldsymbol{x}) + \underbrace{\nabla_\theta f_{\theta_0}(\boldsymbol{x})^\top}_{\text{Tangent feature}} \Delta\theta_T, \tag{3}$$

*where $\Delta\theta_T = \theta_T - \theta_0$ denotes the parameter deviation throughout t training iterations.*

The NTK regime is first studied using infinite-wide neural networks (Jacot et al., 2018; Lee et al., 2019) and is later applied to model fine-tuning (Malladi et al., 2023). In active adaptation, we only continuously train a model for a few epochs, matching the model fine-tuning setting (Malladi et al., 2023). Under Assumption 3, it is easy to see that the prediction variability at a given data sample $\boldsymbol{x}$ throughout the training procedure is upper bounded by its tangent feature norm $\|\nabla_\theta f_{\theta_0}(\boldsymbol{x})\|$ and the magnitude of parameter derivation:

$$\underbrace{\|f_{\theta_T}^{\mathrm{lin}}(\boldsymbol{x}) - f_{\theta_0}(\boldsymbol{x})\|}_{\text{Prediction variability}} \leq \underbrace{\underbrace{\|\nabla_\theta f_{\theta_0}(\boldsymbol{x})\|}_{\text{Feature norm}} \|\Delta\theta_T\|}_{\text{Variability upper bound}} . \tag{4}$$

Although the model is linearized, the loss function remains non-linear. We further introduce an upper bound of the loss reduction gap that is composed of (1) a term that is associated with a gradient distance and (2) a variability upper bound based on a maximum feature norm.

**Theorem 4.** *(Loss reduction gap upper bound) Let $\ell(\boldsymbol{x}, y : f_\theta) = \|f_\theta(\boldsymbol{x}) - y\|^2$ be a mean square error (MSE) loss function, with definitions in Section 3 and Assumption 3, we have:*

$$r_{0\to T}(\boldsymbol{x}, y; f_\theta^{\mathrm{lin}}) - r_{0\to T}(\boldsymbol{x}', y'; f_\theta^{\mathrm{lin}}) \leq \underbrace{\|\nabla_\theta\ell(\boldsymbol{x}, y; f_{\theta_0}) - \nabla_\theta\ell(\boldsymbol{x}', y'; f_{\theta_0})\|}_{\text{Gradient distance}} \|\Delta\theta_T\|$$
$$+ 2\underbrace{\max(\|\nabla_\theta f_{\theta_0}(\boldsymbol{x})\|, \|\nabla_\theta f_{\theta_0}(\boldsymbol{x}')\|)^2}_{\text{Max feature norm}} \|\Delta\theta_T\|^2 . \tag{5}$$
$$\underbrace{\phantom{+ 2\max(\|\nabla_\theta f_{\theta_0}(\boldsymbol{x})\|, \|\nabla_\theta f_{\theta_0}(\boldsymbol{x}')\|)^2 \|\Delta\theta_T\|^2}}_{\text{Variability upper bound}}$$

The upper bound in Equation 5 is easy to interpret: (1) the first term with gradient distance captures a first-order similarity between loss reductions of $\boldsymbol{x}$ and $\boldsymbol{x}'$ and (2) the variability upper bound indicates the difficulty of maintaining a higher-order loss reduction similarity during many training iterations. An interesting observation is that the high-order variability term relies on the square of a first-order tangent feature instead of high-order gradients. Figure 2 illustrates the main result.

**Implications.** The gradient distance term in our main result complements previous works on active learning with gradient embedding (Ash et al., 2020). In addition, the variability upper bound term further implies that minimizing the gradient distance can be insufficient when adaptation training contains many iterations. Without explicitly considering the variability term, an active learning algorithm may neglect the data samples that are very difficult to learn by learning its close neighbors. According to Proposition 2, a large loss reduction gap may result in diminished expected loss reduction. To this end, we will present an improved approach that considers the prediction variability and improves the single-round active adaptation with many training iterations in the adaptation procedure.

## 5 Approach

The goal of our algorithm design is to avoid neglecting the high-variability data samples so that we can directly maximize the training loss reduction over them while keeping a small loss reduction gap small (Proposition 2 in Section 4). In a bird's eye view, our approach operates under the conventional two-step active adaptation framework (Section 3.2), where the first step is to represent each data sample by its embedding and then conduct data acquisition (e.g., k-center greedy) in an embedding space. This framework is similar to prior

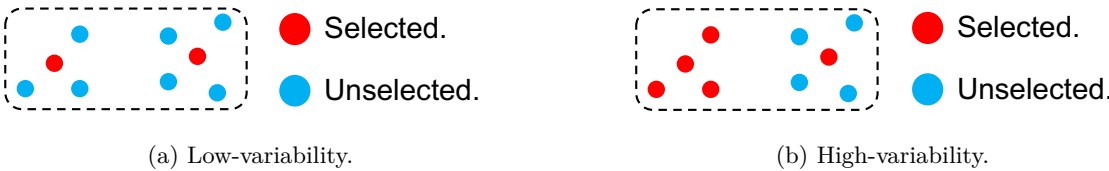

(a) Low-variability.                    (b) High-variability.

Figure 3: When data samples have low-variability (a), selecting one data sample can be sufficient to represent a cluster of data samples. However, in a high-variability setting (b), data samples can hardly represent each other, even if they are close (e.g., the left cluster with red dots).

work in terms of using gradient embedding in the first step but also differs in the sense that it adds a variation to the gradient embedding to encourage the direct selection and learning of high-variability samples, which may lead to a large loss reduction gap, following our theoretical analysis in Section 4. Figure 3 provides an intuitive example where we select all the high-variability samples from one cluster and selectively pick a few representative low-variability samples from other clusters. We also derive efficient implementations for various tasks, including image and text classification, question-answering, and reward modeling.

### 5.1 Feature-norm Scaled Gradient Embedding (FORGE)

In the previous section, the theoretical analysis shows that achieving a good active adaptation performance requires minimizing a loss reduction gap (Theorem 2), which further depends on a gradient distance between a pair of samples and their variability upper bound (Theorem 4). As is detailed in the preliminary (Section 3), we perform diversity-based data selection in a gradient space, i.e., $\phi(\boldsymbol{x}, y, f_{\theta_0}, \ell) = \nabla_\theta \ell(\boldsymbol{x}, y; f_{\theta_0})$, which encourage a k-center greedy algorithm (Section 3.2) effectively minimizes the gradient distance term, $\|\nabla_\theta \ell(\boldsymbol{x}, y; f_{\theta_0}) - \nabla_\theta \ell(\boldsymbol{x}', y'; f_{\theta_0})\|$. However, diversity sampling over gradient space is insufficient for minimizing the variability upper bound term, $\max(\|\nabla_\theta f_{\theta_0}(\boldsymbol{x})\|, \|\nabla_\theta f_{\theta_0}(\boldsymbol{x}')\|)^2 \|\Delta\theta\|^2$. Neglecting the variability upper bound can lead to sub-optimal selection results because we will miss certain samples with a small gradient distance from their selected neighbors but deviate from their neighbors after many training iterations due to high prediction variability (Figure 3). To achieve a low prediction variability and a small gradient distance simultaneously, we develop a new embedding approach called feature-norm scaled gradient embedding (FORGE):

**Definition 5.** *(FORGE) A feature-norm scaled gradient embedding function $\phi$ is defined as:*

$$\phi(\boldsymbol{x}, y, f_{\theta_0}, \ell) = \underbrace{\frac{\|\nabla_\theta f_{\theta_0}(\boldsymbol{x})\|}{\|\nabla_\theta \ell(\boldsymbol{x}, y; f_{\theta_0})\|}}_{\textit{Feature norm re-scaling}} \cdot \underbrace{\nabla_\theta \ell(\boldsymbol{x}, y; f_{\theta_0})}_{\textit{Gradient embedding}} \tag{6}$$

Our FORGE embedding is a re-scaled version of the gradient embedding $\nabla_\theta \ell(\boldsymbol{x}, y; f_{\theta_0})$, where the feature norm $\|\nabla_\theta f_{\theta_0}(\boldsymbol{x})\|$ is the re-scaling factor and decides the magnitude of the FORGE embedding. Since the feature norm also decides the variability upper bound under unknown training dynamics, we assign high-magnitude embedding vectors to high-variability samples. This strategy is effective because diversity-based data selection methods seek data samples far away from selected ones. High-magnitude vectors are often further away from others compared to low-magnitude vectors, which usually fall into a few clusters around 0. Figure 4 further illustrates the advantage of our FORGE embedding and makes comparisons. By re-scaling the embedding vector according to the prediction variability, we avoid neglecting data samples with a low gradient embedding magnitude but a high prediction variability. We further investigate under which condition our FORGE approach with re-sacling operation can outperform the baseline BADGE approach without re-scaling.

**Theorem 6.** *Let $\gamma_{BADGE}$ and $\gamma_{FORGE}$ be the maximum feature norm of any data sample in $\hat{\mathcal{S}}_{BADGE}$ and $\hat{\mathcal{S}}_{FORGE}$, respectively. $\Gamma_{BADGE}$ is an upper bound of the loss reduction gap in Equation 2, $\max_{\boldsymbol{x}', y', \boldsymbol{x}, y \in \hat{\mathcal{S}}_{BADGE}} |r_{0 \to T}(\boldsymbol{x}, y; f_\theta) - r_{0 \to T}(\boldsymbol{x}', y'; f_\theta)| \le \Gamma_{BADGE}$, and $\Gamma_{FORGE}$ is also an upper bound of $\max_{\boldsymbol{x}', y', \boldsymbol{x}, y \in \hat{\mathcal{S}}_{FORGE}} |r_{0 \to T}(\boldsymbol{x}, y; f_\theta) - r_{0 \to T}(\boldsymbol{x}', y'; f_\theta)|$. If the FORGE embedding helps select large feature norm samples such that $\gamma_{BADGE} > \gamma_{FORGE}$, when the parameter deviation is large such that $\|\Delta\theta_T\| >$*

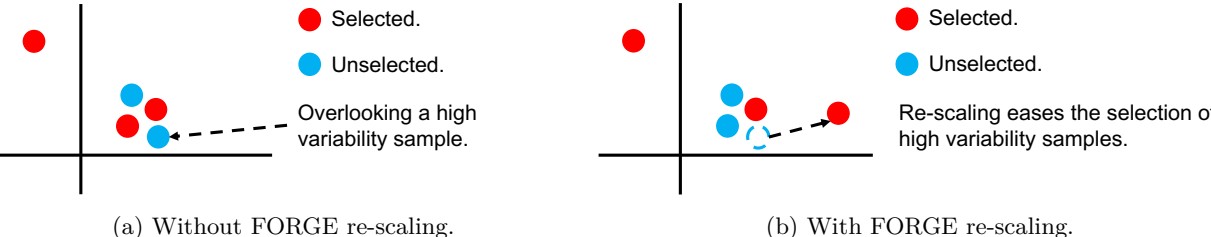

(a) Without FORGE re-scaling.                    (b) With FORGE re-scaling.

Figure 4: The re-scaling operation in FORGE scales gradient embedding according to their prediction variability and intentionally "pops out" high-variability samples in a gradient embedding space. This re-scaling helps diversity-based data acquisition methods (e.g., k-center greedy) pick high-variability samples.

$\frac{(\diamond-1)\cdot\epsilon+\bigcirc}{2(\gamma_{BADGE}-\gamma_{FORGE})}$, *where* $\diamond = \max_{\boldsymbol{x},y\in\mathcal{S}_{FORGE}} \frac{\|\nabla_\theta \ell(\boldsymbol{x},y;f_{\theta_0})\|}{\|\nabla_\theta f_{\theta_0}(\boldsymbol{x})\|}$ *and* $\bigcirc = \max_{\boldsymbol{x}',y',\boldsymbol{x},y\in\hat{\mathcal{S}}_{FORGE}} \left\| \nabla_\theta \ell(\boldsymbol{x}',y';f_{\theta_0}) - \frac{\|\nabla_\theta \ell(\boldsymbol{x},y;f_{\theta_0})\|}{\|\nabla_\theta f_{\theta_0}(\boldsymbol{x})\|} \cdot \frac{\|\nabla_\theta f_{\theta_0}(\boldsymbol{x}')\|}{\|\nabla_\theta \ell(\boldsymbol{x}',y';f_{\theta_0})\|} \cdot \nabla_\theta \ell(\boldsymbol{x}',y';f_{\theta_0}) \right\|$ *are constants, we have*

$$\Gamma_{FORGE} < \Gamma_{BADGE}. \tag{7}$$

This theorem implies that if the re-scaling operation in FORGE embedding can help selecting data samples with large feature norm and making $\gamma_{\text{FORGE}}$ – the maximum feature norm among unselected data samples and their corresponding selected neighbor $\hat{\mathcal{S}}_{FORGE}$ – smaller than $\gamma_{\text{BADGE}}$, the loss reduction gap upper bound $\Gamma_{\text{FORGE}}$ of the FORGE embedding is provably smaller. Combining with Proposition 2 and taking the training loss reduction into consideration, we can further see that if the parameter deviation is large such that

$\|\Delta\theta_T\| > \frac{(\diamond-1)\cdot\epsilon+\bigcirc+\sqrt{[(1-\diamond)\epsilon+\bigcirc]^2-8(\gamma_{\text{BADGE}}-\gamma_{\text{FORGE}})[\sum_{i\in\boldsymbol{s}_{FORGE}} w_i r_{0\to T}(\boldsymbol{x}_i,y_i;f_\theta) - \sum_{i\in\boldsymbol{s}_{BADGE}} w_i r_{0\to T}(\boldsymbol{x}_i,y_i;f_\theta)]}}{4(\gamma_{\text{BADGE}}-\gamma_{\text{FORGE}})}$,

our FORGE approach achieve a higher lower bound of the expected loss reduction.

## 5.2 Efficient Implementation

Computing the tangent feature $\nabla_\theta f_{\theta_0}(\boldsymbol{x})$ and the gradient embedding $\nabla_\theta \ell(\boldsymbol{x},y;f_{\theta_0})$ are expensive. To alleviate the computational overhead, prior work (Ash et al., 2020) showed that using the last layer's gradient embedding of a neural network achieves competitive performance. We extend this result and provide analytical constructions of FORGE embedding using only a single forward pass of a neural network. Such single-pass construction applies to various vision and language tasks. Our analytical construction involves the last hidden state (i.e., the input to the last linear layer) $\boldsymbol{z}$, the sigmoid activation function $\sigma$, and the pseudo-label (i.e., the prediction) $\hat{y}$. The pseudo-label $\hat{y}$ is a common surrogate of the true label $y$ (Ash et al., 2020), which is not yet available in the data selection step. Notably, all the following analytical constructions require only a forward pass through the neural network and are, therefore. computationally efficient.

**Classification task.**   In a binary classification task, we have $\phi(\boldsymbol{x},\hat{y},f_{\theta_0},\boldsymbol{z}) = \frac{\|\boldsymbol{z}\|}{\|\sigma(f(\boldsymbol{x})-\hat{y})\boldsymbol{z}\|} \cdot \sigma(f_{\theta_0}(\boldsymbol{x})-\hat{y})\boldsymbol{z}$. In sequence classification tasks where each input token has a corresponding hidden state, we use the hidden state $\boldsymbol{z}_{\text{CLS}}$ of the [CLS] token. Concatenating the FORGE embedding vector of each class extends our approach to multi-class cases.

**Span-based QA task.**   Span-based question-answering (QA) task requires a model to predict the starting index $\hat{y}_s$ and the end index $\hat{y}_e$ of an answer in a sequence, using two separated linear layers. For a given sequence with $L$ tokens, we have $\phi(\boldsymbol{z}_s,\hat{y}_s,f_{\theta_0,s}) = \frac{1}{L}\sum_{i=1}^L \frac{\|\boldsymbol{z}_{s,i}\|}{\|\sigma(f_{\theta_0,s}(\boldsymbol{x})_i - \hat{y}_{s,i})\boldsymbol{z}_{s,i}\|} \cdot \sigma(f_{s,\theta_0}(\boldsymbol{x})_{s,i} - \hat{y}_{s,i})\boldsymbol{z}_{s,i}$ Then, we concatenate the staring and ending FORGE embeddings: $\text{Cat}\Big(\phi(\boldsymbol{x},\hat{y}_s,f_{\theta_0,s},\boldsymbol{z}_s), \phi(\boldsymbol{x},\hat{y}_e,f_{\theta_0,e},\boldsymbol{z}_e)\Big)$

**Reward modeling task.**   The loss function of a reward modeling task is $\ell(\boldsymbol{x}^{\text{w}},\boldsymbol{x}^{\text{l}},f_\theta) = \log \sigma\Big(f_\theta(\boldsymbol{x}^{\text{w}}) - f_\theta(\boldsymbol{x}^{\text{l}})\Big)$, where $\boldsymbol{x}^{\text{w}}$ is the preferred winning sample and $\boldsymbol{x}^{\text{l}}$ is the other loss sample. Note that the subtraction

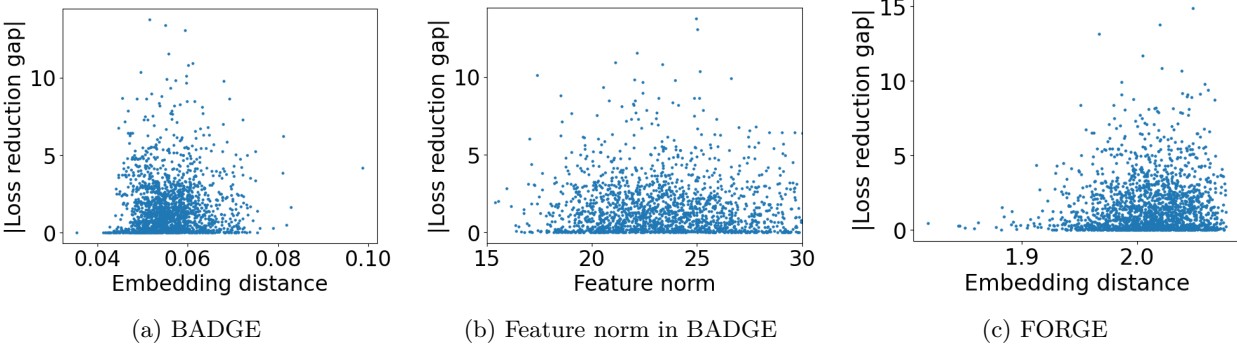

Figure 5: The loss reduction gap can be large even if the gradient embedding (BADGE) distance is small (a). The feature norm explains the large loss reduction gap (b). Incorporating the feature norm term in FORGE embedding alleviates this issue (c).

$f_\theta(\boldsymbol{x}^{\mathrm{w}}) - f_\theta(\boldsymbol{x}^{\mathrm{l}})$ within the sigmoid function $\sigma$ equals to $\boldsymbol{z}^{\mathrm{w}\top}\theta^{-1} - \boldsymbol{z}^{\mathrm{l}\top}\theta^{-1}$, where $\theta^{-1}$ is the parameter of the last linear layer. Therefore, we can consider $\boldsymbol{z}^{\mathrm{w}} - \boldsymbol{z}^{\mathrm{l}}$ as the input to the last layer, ignore the gradient $(e^{\boldsymbol{z}^{\mathrm{w}\top}\theta^{-1} - \boldsymbol{z}^{\mathrm{l}\top}\theta^{-1}} + 1)^{-1}$ of the log-sigmoid function because it is a positive scalar, and have $\phi(\boldsymbol{z}^{\mathrm{w}}, \boldsymbol{z}^{\mathrm{l}}) = \boldsymbol{z}^{\mathrm{w}} - \boldsymbol{z}^{\mathrm{l}}$.

## 6 Experiments

We present the empirical verification of our approach and make comparisons with strong baselines.

### 6.1 Setup

**Tasks and datasets.** In the image classification task, we use the VLCS dataset (Gulrajani & Lopez-Paz, 2021) and the VisDA dataset (Peng et al., 2017). The sentiment classification task operates over the Amazon and Yelp review datasets (McAuley et al., 2015; Zhang et al., 2015). The span-based question-answering (QA) task employs the Squad and News datasets (Rajpurkar et al., 2016; Trischler et al., 2017). The reward modeling task utilizes the Anthropic-hh-rlhf dataset (Bai et al., 2022). We split the data evenly across sources during training, which consistently stabilizes training. For example, in VLCS (4 domains), when the source domain is Caltech and the total batch size is 16, we allocate 4 samples to the source domain in each batch.

**Model architecture.** We use the Resnet-50 model (He et al., 2015) for the image classification task. The sentiment classification and the span-based QA task adopt the distilled-Bert models (Devlin et al., 2019; Sanh et al., 2019) with a classification head and a QA head, respectively. We consider the GPT-2-medium model (Radford et al., 2019) for reward modeling tasks.

**Hyper-parameters.** We use the SGD optimization for the Resnet-50 model, the Adam optimizer for the distilled-Bert models, and the GPT-2 model. The initial learning rate is 1e-4 for all adaptation tasks, and we use linear decay scheduling for the GPT-2 model in a reward modeling task. The number of epochs for adaptation tasks is 4, and we train the reward model for 1 epoch. The batch size for the Resnet-50 model is 64, the distilled-Bert model is 16, and the GPT-2 model is 4.

### 6.2 Baselines

We include random selection, uncertainty-based selection: margin (Balcan et al., 2007) and DUC (Xie et al., 2023), diversity-based selection: CORESET (Sener & Savarese, 2018), BADGE (Ash et al., 2020), TypiClust (Hacohen et al., 2022), and ProbCover (Yehuda et al., 2022), and hybrid selection: CLUE (Prabhu et al., 2021) and BiLAF (Lu et al., 2024). Approaches such as DynamicAL (Wang et al., 2022) and DULO (Wang et al., 2023) are omitted due to their prohibitive computational overhead as one requires computing neural tangent kernel, and the other needs training 4000 proxy models.

Table 1: Accuracy decomposition of active adaptation methods with 5% labels.

| Method | Selected Train | Unselected Train | Validation | Test |
|---|---|---|---|---|
| Random | $100.0_{\pm 0.00}$ | $74.30_{\pm 0.71}$ | $74.52_{\pm 0.65}$ | $73.89_{\pm 0.76}$ |
| CORESET | $100.0_{\pm 0.00}$ | $73.68_{\pm 0.08}$ | $73.67_{\pm 0.25}$ | $72.99_{\pm 0.16}$ |
| BADGE | $100.0_{\pm 0.00}$ | $75.18_{\pm 0.22}$ | $74.15_{\pm 0.33}$ | $73.28_{\pm 0.21}$ |
| CLUE | $100.0_{\pm 0.00}$ | $75.90_{\pm 0.36}$ | $73.77_{\pm 0.33}$ | $72.97_{\pm 0.15}$ |
| FORGE (ours) | $100.0_{\pm 0.00}$ | $\mathbf{76.13}_{\pm \mathbf{0.06}}$ | $\mathbf{75.19}_{\pm 0.26}$ | $\mathbf{74.94}_{\pm \mathbf{0.08}}$ |

Note: The numbers are average accuracy over three runs. Variance is rounded up.

## 6.3 Visualizing Loss Reduction Gap

We investigate the correlation between the embedding distance and the absolute loss reduction gap in Proposition 2. This correlation is important because diversity-based data selection methods aim to minimize the distance between pairs on selected and unselected samples (lines 3-6, Algorithm 1).

In these experiments, we use the Caltech (C) and VOC (V) datasets from the VLCS dataset with 5 classes. We first fine-tune a pre-trained Resnet-50 on the Caltech dataset to obtain a source model. Then, we use CORESET, BADGE, and our approaches to select 5% data samples from the VOC dataset (i.e., target domain). The source model is further fine-tuned on the target domain to get the final accuracy, where each batch is split evenly for source and target data. For the VLCS dataset, we report the average accuracy on the V, L, and S datasets. We always find using source data stabilizes and improves adaptation.

In Figure 5, we first plot the correlation between the gradient distance between each unselected sample $\boldsymbol{x}'$ and its closest select neighbor $\boldsymbol{x}$, $\|\phi_{\text{BADGE}}(\boldsymbol{x}, y, f_{\theta_0}, \ell) - \phi_{\text{BADGE}}(\boldsymbol{x}', y', f_{\theta_0}, \ell)\|$, and their absolute loss reduction gap, $|r_{0 \to K}(\boldsymbol{x}, y; f_\theta) - r_{0 \to K}(\boldsymbol{x}', y'; f_\theta)|$. With the gradient embedding (BADGE), the loss reduction gap significantly increases with a minor increase in the embedding distance, diminishing their correlation and hurting the diversity-based selection performance. Then, we show that the diminished correlation can be explained by the (tangent) feature norm in Figure 5b. Adopting FORGE, which explicitly considers prediction variability via the feature norm term, recovers a strong correlation between the embedding distance and the loss reduction gap.

Table 2: Accuracy of active adaptation methods.

| Method | Image-CLS on VLCS | | Image-CLS on VisDA | | Average |
|---|---|---|---|---|---|
| | 5% | 10% | 5% | 10% | |
| Random | $74.87_{\pm 0.74}$ | $75.33_{\pm 0.76}$ | $81.91_{\pm 0.01}$ | $84.28_{\pm 0.01}$ | 79.10 |
| Margin | $61.43_{\pm 0.17}$ | $64.32_{\pm 0.18}$ | $81.08_{\pm 0.01}$ | $83.59_{\pm 0.01}$ | 72.61 |
| DUC | $68.07_{\pm 0.15}$ | $72.78._{\pm 0.05}$ | $81.59_{\pm 0.01}$ | $85.20_{\pm 0.02}$ | 76.91 |
| CORESET | $73.50_{\pm 0.06}$ | $74.45_{\pm 0.26}$ | $81.04_{\pm 0.01}$ | $84.85_{\pm 0.01}$ | 78.46 |
| BADGE | $73.74_{\pm 0.18}$ | $74.82_{\pm 1.71}$ | $81.37_{\pm 0.01}$ | $84.54_{\pm 0.01}$ | 78.62 |
| CLUE | $74.56_{\pm 0.41}$ | $75.56_{\pm 0.08}$ | $81.42_{\pm 0.02}$ | $84.82_{\pm 0.02}$ | 79.09 |
| TypiClust | $74.94_{\pm 0.59}$ | $75.23_{\pm 1.33}$ | $81.75_{\pm 0.68}$ | $85.05_{\pm 0.81}$ | 79.24 |
| ProbCover | $73.26_{\pm 0.52}$ | $73.69_{\pm 0.70}$ | $81.55_{\pm 0.66}$ | $84.88_{\pm 0.34}$ | 78.35 |
| BiLAF | $74.87_{\pm 0.26}$ | $75.02_{\pm 0.33}$ | $81.38_{\pm 0.57}$ | $84.44_{\pm 0.49}$ | 78.93 |
| FORGE (ours) | $\mathbf{75.19}_{\pm \mathbf{0.01}}$ | $\mathbf{76.06}_{\pm \mathbf{0.07}}$ | $\mathbf{82.28}_{\pm \mathbf{0.01}}$ | $\mathbf{85.45}_{\pm \mathbf{0.01}}$ | $\mathbf{79.75}$ |

Note: The numbers are average accuracy over three runs. Variance is rounded up.

Table 3: Accuracy of active adaptation methods.

| Method | Sentiment-CLS | | Span-QA | | Average |
| | 5% | 10% | 5% | 10% | |
|---|---|---|---|---|---|
| Random | $50.53_{\pm 0.01}$ | $51.66_{\pm 0.01}$ | $38.27_{\pm 0.25}$ | $38.84_{\pm 0.04}$ | 44.83 |
| Margin | $44.41_{\pm 0.02}$ | $48.57_{\pm 0.01}$ | $33.01_{\pm 0.11}$ | $35.92_{\pm 0.14}$ | 30.48 |
| DUC | $48.21_{\pm 0.01}$ | $50.66_{\pm 0.01}$ | $33.59_{\pm 0.21}$ | $37.82_{\pm 0.09}$ | 42.57 |
| CORESET | $51.34_{\pm 0.01}$ | $51.33_{\pm 0.01}$ | $37.57_{\pm 0.17}$ | $38.68_{\pm 0.02}$ | 44.73 |
| BADGE | $51.26_{\pm 0.01}$ | $\mathbf{52.28}_{\pm 0.02}$ | $38.25_{\pm 0.09}$ | $38.91_{\pm 0.07}$ | 45.18 |
| CLUE | $50.90_{\pm 0.01}$ | $51.65_{\pm 0.01}$ | $38.21_{\pm 0.09}$ | $38.45_{\pm 0.06}$ | 44.80 |
| TypiClust | $51.38_{\pm 0.02}$ | $51.94_{\pm 0.02}$ | $38.38_{\pm 0.01}$ | $38.87_{\pm 0.01}$ | 45.14 |
| ProbCover | $50.90_{\pm 0.02}$ | $51.58_{\pm 0.01}$ | $38.19_{\pm 0.02}$ | $38.41_{\pm 0.03}$ | 44.77 |
| BiLAF | $50.47_{\pm 0.03}$ | $50.78_{\pm 0.01}$ | $37.94_{\pm 0.02}$ | $38.12_{\pm 0.02}$ | 44.31 |
| FORGE (ours) | $\mathbf{51.79}_{\pm 0.01}$ | $52.23_{\pm 0.01}$ | $\mathbf{38.67}_{\pm 0.13}$ | $\mathbf{39.06}_{\pm 0.18}$ | $\mathbf{45.44}$ |

Note: The numbers are average accuracy over three runs. Variance is rounded up.

## 6.4 Performance Evaluation

We further show that FORGE, which recovers a strong correlation between the embedding distance and the loss reduction gap (Section 6.3), improves active adaptation performance. Table 1 lists the accuracy decomposition of active adaptation methods in an image classification task (C to V adaptation in VLCS) with a labeling budget of 5%. Our approach improves the accuracy on the unselected training set and achieves a better test performance. Such an advantage further extends to another image classification (C to VLS), sentiment classification, question answering, and reward modeling tasks with labeling budgets of 5%, 10%, and 20%.

In the image classification experiments, we consider a more challenging setting. The Caltech (C) dataset remains the source domain, and we use a mix of the VOC (V), LabelME (L), and SUN (S) as the target domains. We select data samples evenly from each target domain (e.g., 5% from each target domain). In VisDA, we consider the synthetic to real transfer. The source domain for sentiment classification is Amazon review, and the target domain is Yelp review. Both datasets have 5 sentiment classes. For the span-QA task, we directly use a fine-tuned distilled-Bert on the Squad dataset [1] and use News as the target domain. The adaptation procedures follow the previous loss reduction gap experiment and always include source domain data in target domain adaptation. We report the model performance on target domains. For the QA task, we report the exact match performance. Our approach achieves the best performance among 7 out of the 8 settings and beats the random selection baseline across all settings. In the sentiment classification task with 5% labeling budget, we observe our approach achieves a 1.26% higher accuracy than the random selection baseline. In contrast, the BADGE approach without the re-scaling operation only beats the random selection baseline in 4 out of the 8 settings.

In the reward modeling task, we first fine-tune a GPT-2-medium model on 50% of the Anthropic-hh-rlhf dataset using the "chosen" response. Then, we employ the warmup strategy in LESS (Xia et al., 2024) and train the supervised fine-tuned (SFT) model using 5% of the remaining data samples, aiming to help the SFT model capture better sentence-level data representations instead of the token-level data representations. We only use the SFT model with warmup in the data selection step. The reward model training starts with the SFT model without any warm-up using the Anthropic-hh-rlhf dataset. We report the accuracy of a hold-out test set. The accuracy of a reward model is measured by the percentage of

Table 4: Reward model accuracy.

| Method | Reward Modeling 20% |
|---|---|
| Random | $63.01_{\pm 0.21}$ |
| CORESET | $63.89_{\pm 0.19}$ |
| BADGE | $63.12_{\pm 0.24}$ |
| FORGE (ours) | $\mathbf{64.45}_{\pm 0.14}$ |

---

[1]https://huggingface.co/distilbert/distilbert-base-cased-distilled-squad

the preferred score being greater than the unpreferred score, $f_\theta(\boldsymbol{x}^{\mathrm{w}}) > f_\theta(\boldsymbol{x}^{\mathrm{l}})$. On average of three different runs, we find that our approach outperform the BADGE approach, which was the second best approach in language tasks (Table 3), by 1.33%.

# 7 Ablation Study

We conduct an ablation study to evaluate the performance of our approach under metrics other than exact match. For the QA benchmark, we report the F1 score in Table 5 and show that our model remains the top performer. Another ablation study (Table 6) on VLCS dataset includes the norm scaling factor in our FORGE embedding construction, considering the $0$, $\frac{1}{2}$, $1$, and $2$ powers of $\frac{\|\nabla_\theta f_{\theta_0}(\boldsymbol{x})\|}{\|\nabla_\theta \ell(\boldsymbol{x},y;f_{\theta_0})\|}$ in Equation 6.

Table 5: F1 score of adaptation methods.

| Method | Sentiment-CLS | |
|---|---|---|
| | 5% | 10% |
| Random | $54.03_{\pm 0.07}$ | $54.73_{\pm 0.14}$ |
| CORESET | $53.08_{\pm 0.17}$ | $54.15_{\pm 0.01}$ |
| BADGE | $54.12_{\pm 0.18}$ | $54.70_{\pm 0.07}$ |
| FORGE (ours) | $\mathbf{54.20}_{\pm \mathbf{0.04}}$ | $\mathbf{54.79}_{\pm \mathbf{0.04}}$ |

Table 6: Accuracy of norm scaling.

| Method | Image-CLS on VLCS | |
|---|---|---|
| | 5% | 10% |
| $\mathrm{norm}^0$ | $73.01_{\pm 0.15}$ | $74.91_{\pm 0.09}$ |
| $\mathrm{norm}^{\frac{1}{2}}$ | $74.63_{\pm 0.17}$ | $75.16_{\pm 0.07}$ |
| $\mathrm{norm}^1$ | $\mathbf{75.19}_{\pm \mathbf{0.01}}$ | $\mathbf{76.06}_{\pm \mathbf{0.07}}$ |
| $\mathrm{norm}^2$ | $74.25_{\pm 0.25}$ | $75.83_{\pm 0.13}$ |

# 8 Conclusion and Future Work

This work studies a single-round active adaptation problem, aiming to reduce the annotation turnaround time and promote timely adaptation to distribution shifts. A single-round adaptation problem requires selecting a subset of data samples for many training iterations. Through theoretical analysis, we show that selecting for many iterations requires considering the prediction variability of each data sample, which is highly correlated with a tangent feature norm. Then, we introduce an improved approach called feature-norm scaled gradient embedding (FORGE) that incorporates prediction variability into the data selection process. Extensive empirical results with various vision and language tasks demonstrate the effectiveness of our approach.

In the future, it would be interesting to study the prediction variability in the pre-training stage (Chen et al., 2023; Tirumala et al., 2024) and the pseudo-label bias in gradient embedding construction in the context of learning with AI feedback (Taori & Hashimoto, 2023; Panickssery et al., 2024).

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

# A  Table of Notations

Table 7: Table of Notations

| Symbol | Description |
|---|---|
| $\boldsymbol{x}, y$ | A data sample |
| $N$ | The number of data samples |
| $\boldsymbol{s}$ | A selected subset of data samples |
| $\theta$ | The parameters of a model |
| $\theta_t$ | The parameters of a model at the $t^{\text{th}}$ training iteration |
| $T$ | The number of training iterations |
| $r_{0\to T}(\boldsymbol{x}, y; f_\theta)$ | The loss reduction $\ell(\boldsymbol{x}, y; f_{\theta_0}) - \ell(\boldsymbol{x}, y; f_{\theta_T})$ after $T$ training iterations |
| $\|\cdot\|$ | The $L_2$ norm of a vector |
| $|\cdot|$ | The size of a set |
| $[N]$ | A set of $N$ natural numbers |
| $\phi$ | An embedding function |
| $\text{Cat}(\cdot, \cdot)$ | A vector concatenation operator |
| $\mathcal{S}$ | A set of selected data samples $\{\boldsymbol{x}_i, y_i \mid i \in \boldsymbol{s}\}$ |
| $\mathcal{S}'$ | A set of unselected data samples $\{\boldsymbol{x}_i, y_i \mid i \in [n] \setminus \boldsymbol{s}\}$ |
| $\hat{\mathcal{S}}$ | A set of unselected data samples and its closet selected neighbor |
| | $\{\boldsymbol{x}_i, y_i, \boldsymbol{x}_j, y_j \mid i \in [n] \setminus \boldsymbol{s}, j = \arg\min_{\boldsymbol{s}} \|\phi(\boldsymbol{x}_i) - \phi(\boldsymbol{x}_j)\|\}$ |
| $\gamma$ | The maximum feature norm, $\max(\|\nabla_\theta f_{\theta_0}(\boldsymbol{x})\|, \|\nabla_\theta f_{\theta_0}(\boldsymbol{x}')\|)$ |

# B  Proofs

**Proposition 2.** *(Decomposition of expected loss reduction) Let $\boldsymbol{x}$ be the closest selected neighbor of an unselected $\boldsymbol{x}'$ and $w_i = c_i + 1$ where $c_i$ is the frequency of each $\boldsymbol{x}$ appears as the closest neighbor, the expected loss reduction $\mathbb{E}_{\mathcal{D}}[r_{0\to T}(\boldsymbol{x}, y; f_\theta)]$ with a given data distribution $\mathcal{D}$ can be decomposed and upper bounded by (1) a training loss reduction, (2) a maximum loss reduction gap, and (3) a generalization gap:*

$$
\mathbb{E}_{\mathcal{D}}[r_{0\to T}(\boldsymbol{x}, y; f_\theta)]
$$

$$
\geq \underbrace{\frac{1}{N}\sum_{i \in \boldsymbol{s}} w_i r_{0\to T}(\boldsymbol{x}_i, y_i; f_\theta)}_{\textit{Weighted training loss reduction}} + \underbrace{\mathbb{E}_{\mathcal{D}}[r_{0\to T}(\boldsymbol{x}, y; f_\theta)] - \frac{1}{N}\sum_{i=1}^{N} r_{0\to T}(\boldsymbol{x}_i, y_i; f)}_{\textit{Generalization gap}}
$$

$$
- \frac{1}{N} \cdot \underbrace{\max_{\boldsymbol{x}', y', \boldsymbol{x}, y \in \hat{\mathcal{S}}} |r_{0\to T}(\boldsymbol{x}, y; f_\theta) - r_{0\to T}(\boldsymbol{x}', y'; f_\theta)|.}_{\textit{Maximum loss reduction gap}}
$$

(8)

*Proof.* Let $(\boldsymbol{x}_{s,i}, y_{s,i})$ be the closest selected neighbor of an unselected $(\boldsymbol{x}_i, y_i)$, when $i \in [N] \setminus \boldsymbol{s}$, and $c_i$ be the frequency of each $(\boldsymbol{x}_i, y_i)$ appears as a closest selected neighbor, we have:

$$
\begin{aligned}
\mathbb{E}_{\mathcal{D}}[r_{0 \to T}(\boldsymbol{x}, y; f)] = {} & \mathbb{E}_{\mathcal{D}}[r_{0 \to T}(\boldsymbol{x}, y; f)] - \frac{1}{N} \sum_{i=1}^{N} r_{0 \to T}(\boldsymbol{x}_i, y_i; f) \\
& + \frac{1}{N} \Big( \sum_{i \in [N] \setminus \boldsymbol{s}} r_{0 \to T}(\boldsymbol{x}_i, y_i; f) + \sum_{i \in \boldsymbol{s}} r_{0 \to T}(\boldsymbol{x}_i, y_i; f) \Big) \\
= {} & \mathbb{E}_{\mathcal{D}}[r_{0 \to T}(\boldsymbol{x}, y; f)] - \frac{1}{N} \sum_{i=1}^{N} r_{0 \to T}(\boldsymbol{x}_i, y_i; f) \\
& + \frac{1}{N} \sum_{i \in [N] \setminus \boldsymbol{s}} \Big( r_{0 \to T}(\boldsymbol{x}_i, y_i; f) - r_{0 \to T}(\boldsymbol{x}_{s,i}, y_{s,i}; f) \Big) \\
& + \frac{1}{N} \sum_{i \in \boldsymbol{s}} r_{0 \to T}(\boldsymbol{x}_i, y_i; f) + \frac{1}{N} \sum_{i \in \boldsymbol{s}} c_i r_{0 \to T}(\boldsymbol{x}_i, y_i; f) \\
= {} & \frac{1}{N} \sum_{i \in \boldsymbol{s}} (c_i + 1) r_{0 \to T}(\boldsymbol{x}_i, y_i; f) + \mathbb{E}_{\mathcal{D}}[r_{0 \to T}(\boldsymbol{x}, y; f)] - \frac{1}{N} \sum_{i=1}^{N} r_{0 \to T}(\boldsymbol{x}_i, y_i; f) \\
& + \frac{1}{N} \sum_{i \in [N] \setminus \boldsymbol{s}} \Big( r_{0 \to T}(\boldsymbol{x}_i, y_i; f) - r_{0 \to T}(\boldsymbol{x}_{s,i}, y_{s,i}; f) \Big) \\
\geq {} & \frac{1}{N} \sum_{i \in \boldsymbol{s}} w_i r_{0 \to T}(\boldsymbol{x}_i, y_i; f) + \mathbb{E}_{\mathcal{D}}[r_{0 \to T}(\boldsymbol{x}, y; f)] - \frac{1}{N} \sum_{i=1}^{N} r_{0 \to T}(\boldsymbol{x}_i, y_i; f) \\
& - \frac{1}{N} \cdot \max_{\boldsymbol{x}', y', \boldsymbol{x}, y \in \hat{\mathcal{S}}} |r_{0 \to T}(\boldsymbol{x}, y; f_\theta) - r_{0 \to T}(\boldsymbol{x}', y'; f_\theta)|.
\end{aligned}
\tag{9}
$$

$\square$

**Theorem 4.** *(Loss reduction gap upper bound) Let $\ell(\boldsymbol{x}, y : f_\theta) = \|f_\theta(\boldsymbol{x}) - y\|^2$ be a mean square error (MSE) loss function, with definitions in Section 3 and Assumption 3, we have:*

$$
r_{0 \to T}(\boldsymbol{x}, y; f_\theta^{\mathrm{lin}}) - r_{0 \to T}(\boldsymbol{x}', y'; f_\theta^{\mathrm{lin}}) \leq \underbrace{\|\nabla_\theta \ell(\boldsymbol{x}, y; f_{\theta_0}) - \nabla_\theta \ell(\boldsymbol{x}', y'; f_{\theta_0})\|}_{\textit{Gradient distance}} \|\Delta \theta_T\|
$$

$$
+ 2 \underbrace{\max(\|\nabla_\theta f_{\theta_0}(\boldsymbol{x})\|, \|\nabla_\theta f_{\theta_0}(\boldsymbol{x}')\|)^2}_{\textit{Max feature norm}} \|\Delta \theta_T\|^2.
\tag{10}
$$

$$
\underbrace{\phantom{+ 2 \max(\|\nabla_\theta f_{\theta_0}(\boldsymbol{x})\|, \|\nabla_\theta f_{\theta_0}(\boldsymbol{x}')\|)^2 \|\Delta \theta_T\|^2}}_{\textit{Variability upper bound}}
$$

*Proof.* Capturing the loss reduction $\ell(\boldsymbol{x}, y; f_{\theta_0}) - \ell(\boldsymbol{x}, y; f_{\theta_T}^{\mathrm{lin}})$ during training is non-trivial because the loss function $\ell$ remains non-linear even if the model $f_{\theta_0}^{\mathrm{lin}}$ is linearized. Therefore, we resort to the Lagrange mean value theorem and show that the loss reduction depends on an interpolated gradient:

$$
r_{0 \to T}(\boldsymbol{x}, y; f_\theta^{\mathrm{lin}}) = \ell(\boldsymbol{x}, y; f_{\theta_0}) - \ell(\boldsymbol{x}, y; f_{\theta_T}^{\mathrm{lin}}) = \nabla_\theta \ell(\boldsymbol{x}, y; f_{\theta_{T,\alpha}}^{\mathrm{lin}})^\top \Delta \theta_T,
\tag{11}
$$

where $f_{\theta_{T,\alpha}}^{\mathrm{lin}} = f_{\theta_0}(\boldsymbol{x}) + \nabla_\theta f_{\theta_0}(\boldsymbol{x}) \cdot \alpha \cdot \Delta \theta_T$ and $\alpha \in [0, 1]$ is an interpolatoin factor. With some re-arrangement of terms, we quantify the deviation between the interpolated gradient and the gradient embedding, which depends on the interpolation factor $\alpha$ and the tangent feature $\nabla_\theta f_{\theta_0}(\boldsymbol{x})$:

$$
\nabla_\theta \ell(f_{\theta_{T,\alpha}}^{\mathrm{lin}}(\boldsymbol{x}), y) = \underbrace{(f_{\theta_0}(\boldsymbol{x}) - y) \nabla_\theta f_{\theta_0}(\boldsymbol{x})}_{\textit{Gradient embedding}} + \underbrace{\alpha \cdot \nabla_\theta f_{\theta_0}(\boldsymbol{x})^\top \Delta \theta_T \nabla_\theta f_{\theta_0}(\boldsymbol{x})}_{\textit{Interpolation deviation}}.
\tag{12}
$$

With this analysis of gradient deviation and the abbreviation $\mathcal{Q}_{\boldsymbol{x},T,\boldsymbol{x}} = \nabla_\theta f_{\theta_0}(\boldsymbol{x})^\top \Delta\theta_T \nabla_\theta f_{\theta_0}(\boldsymbol{x})$, we have the following upper bound on pair-wise interpolated gradient distance:

$$
\begin{aligned}
&\|\nabla_\theta \ell(\boldsymbol{x}, y; f^{\text{lin}}_{\theta_{T,\alpha}}) - \nabla_\theta \ell(\boldsymbol{x}', y'; f^{\text{lin}}_{\theta_{T,\alpha'}})\| \\
&\leq \underbrace{\|(f_{\theta_0}(\boldsymbol{x}) - y)\nabla_\theta f_{\theta_0}(\boldsymbol{x}) - (f_{\theta_0}(\boldsymbol{x}') - y')\nabla_\theta f_{\theta_0}(\boldsymbol{x}')\|}_{\text{Gradient distance}} \\
&\quad + \underbrace{\|\alpha \cdot \mathcal{Q}_{\boldsymbol{x},T,\boldsymbol{x}} - \alpha' \cdot \mathcal{Q}_{\boldsymbol{x},T,\boldsymbol{x}})\|}_{\text{Interpolation distance}} + \underbrace{\|\alpha' \cdot \mathcal{Q}_{\boldsymbol{x},T,\boldsymbol{x}} - \alpha' \cdot \mathcal{Q}_{\boldsymbol{x}',T,\boldsymbol{x}'})\|}_{\text{Feature distance}},
\end{aligned}
\tag{13}
$$

where the first term quantifies the contribution of gradient embedding. For the latter two terms, we further present upper bounds that grow w.r.t. the feature norms $\|\nabla_\theta f_0(\boldsymbol{x})\|$ or $\|\nabla_\theta f_0(\boldsymbol{x}')\|$. For the interpolation distance, we have:

$$
\|\alpha \cdot \mathcal{Q}_{\boldsymbol{x},T,\boldsymbol{x}} - \alpha' \cdot \mathcal{Q}_{\boldsymbol{x},T,\boldsymbol{x}}\| \leq \|\nabla_\theta f_{\theta_0}(\boldsymbol{x})\|^2 \|\Delta\theta_t\|.
\tag{14}
$$

For the feature distance, we have:

$$
\begin{aligned}
&\|\alpha' \cdot \mathcal{Q}_{\boldsymbol{x},T,\boldsymbol{x}} - \alpha' \cdot \mathcal{Q}_{\boldsymbol{x}',T,\boldsymbol{x}'})\| \\
&\leq \|\nabla_\theta f_{\theta_0}(\boldsymbol{x}) - \nabla_\theta f_{\theta_0}(\boldsymbol{x}')\|\|\Delta\theta_T\| \cdot \max(\|\nabla_\theta f_{\theta_0}(\boldsymbol{x})\|, \|\nabla_\theta f_{\theta_0}(\boldsymbol{x}')\|) \\
&\leq \max(\|\nabla_\theta f_{\theta_0}(\boldsymbol{x})\|, \|\nabla_\theta f_{\theta_0}(\boldsymbol{x}')\|)^2 \|\Delta\theta_T\|.
\end{aligned}
\tag{15}
$$

Plugging Equations 14 and 15 into 13, we have:

$$
\begin{aligned}
&\|\nabla_\theta \ell(\boldsymbol{x}, y; f^{\text{lin}}_{\theta_{T,\alpha}}) - \nabla_\theta \ell(\boldsymbol{x}', y'; f^{\text{lin}}_{\theta_{T,\alpha'}})\| \\
&\leq \|\nabla_\theta \ell(\boldsymbol{x}, y; f^{\text{lin}}_{\theta_0}) - \nabla_\theta \ell(\boldsymbol{x}', y'; f^{\text{lin}}_{\theta_0})\| + 2\max(\|\nabla_\theta f_{\theta_0}(\boldsymbol{x})\|, \|\nabla_\theta f_{\theta_0}(\boldsymbol{x}')\|)^2 \|\Delta\theta_T\|.
\end{aligned}
\tag{16}
$$

Combining Equations 11 and 16, we complete the proof:

$$
\begin{aligned}
&r_{0\to T}(\boldsymbol{x}, y; f^{\text{lin}}_\theta) - r_{0\to T}(\boldsymbol{x}', y'; f^{\text{lin}}_\theta) \\
&= \left(\nabla_\theta \ell(\boldsymbol{x}, y; f^{\text{lin}}_{\theta_{T,\alpha}}) - \nabla_\theta \ell(\boldsymbol{x}', y'; f^{\text{lin}}_{\theta_{T,\alpha'}})\right)^\top \Delta\theta_T \\
&\leq \|\nabla_\theta \ell(\boldsymbol{x}, y; f^{\text{lin}}_{\theta_0}) - \nabla_\theta \ell(\boldsymbol{x}', y'; f^{\text{lin}}_{\theta_0})\|\|\Delta\theta_T\| \\
&\quad + 2\max(\|\nabla_\theta f_{\theta_0}(\boldsymbol{x})\|, \|\nabla_\theta f_{\theta_0}(\boldsymbol{x}')\|)^2 \|\Delta\theta_T\|^2.
\end{aligned}
\tag{17}
$$

$\square$

**Theorem 6.** *Let $\gamma_{BADGE}$ and $\gamma_{FORGE}$ be the maximum feature norm of any data sample in $\hat{\mathcal{S}}_{BADGE}$ and $\hat{\mathcal{S}}_{FORGE}$, respectively. $\Gamma_{BADGE}$ is an upper bound of the loss reduction gap in Equation 2, $\max_{\boldsymbol{x}',y',\boldsymbol{x},y\in\hat{\mathcal{S}}_{BADGE}} |r_{0\to T}(\boldsymbol{x}, y; f_\theta) - r_{0\to T}(\boldsymbol{x}', y'; f_\theta)| \leq \Gamma_{BADGE}$, and $\Gamma_{FORGE}$ is also an upper bound of $\max_{\boldsymbol{x}',y',\boldsymbol{x},y\in\hat{\mathcal{S}}_{FORGE}} |r_{0\to T}(\boldsymbol{x}, y; f_\theta) - r_{0\to T}(\boldsymbol{x}', y'; f_\theta)|$. If the FORGE embedding helps select large feature norm samples such that $\gamma_{BADGE} > \gamma_{FORGE}$, when the parameter deviation is large such that $\|\Delta\theta_T\| > \frac{(\diamond - 1)\cdot\epsilon + \bigcirc}{2(\gamma_{BADGE} - \gamma_{FORGE})}$, where $\diamond = \max_{\boldsymbol{x},y\in\mathcal{S}_{FORGE}} \frac{\|\nabla_\theta \ell(\boldsymbol{x},y;f_{\theta_0})\|}{\|\nabla_\theta f_{\theta_0}(\boldsymbol{x})\|}$ and $\bigcirc = \max_{\boldsymbol{x}',y',\boldsymbol{x},y\in\hat{\mathcal{S}}_{FORGE}} \left\|\nabla_\theta \ell(\boldsymbol{x}',y';f_{\theta_0}) - \frac{\|\nabla_\theta \ell(\boldsymbol{x},y;f_{\theta_0})\|}{\|\nabla_\theta f_{\theta_0}(\boldsymbol{x})\|} \cdot \frac{\|\nabla_\theta f_{\theta_0}(\boldsymbol{x}')\|}{\|\nabla_\theta \ell(\boldsymbol{x}',y';f_{\theta_0})\|} \cdot \nabla_\theta \ell(\boldsymbol{x}',y';f_{\theta_0})\right\|$ are constants, we have*

$$
\Gamma_{FORGE} < \Gamma_{BADGE}.
\tag{18}
$$

*Proof.* By the FORGE embedding definition, we have

$$
\begin{aligned}
&\|\phi(\boldsymbol{x}, y, f_{\theta_0}, \ell) - \phi(\boldsymbol{x}', y', f_{\theta_0}, \ell)\| \\
&= \left\| \frac{\|\nabla_\theta f_{\theta_0}(\boldsymbol{x})\|}{\|\nabla_\theta \ell(\boldsymbol{x}, y; f_{\theta_0})\|} \cdot \nabla_\theta \ell(\boldsymbol{x}, y; f_{\theta_0}) - \frac{\|\nabla_\theta f_{\theta_0}(\boldsymbol{x}')\|}{\|\nabla_\theta \ell(\boldsymbol{x}', y'; f_{\theta_0})\|} \cdot \nabla_\theta \ell(\boldsymbol{x}', y'; f_{\theta_0}) \right\| \\
&= \frac{\|\nabla_\theta f_{\theta_0}(\boldsymbol{x})\|}{\|\nabla_\theta \ell(\boldsymbol{x}, y; f_{\theta_0})\|} \cdot \left\| \nabla_\theta \ell(\boldsymbol{x}, y; f_{\theta_0}) - \nabla_\theta \ell(\boldsymbol{x}', y'; f_{\theta_0}) \right. \\
&\quad\left. + \nabla_\theta \ell(\boldsymbol{x}', y'; f_{\theta_0}) - \frac{\|\nabla_\theta \ell(\boldsymbol{x}, y; f_{\theta_0})\|}{\|\nabla_\theta f_{\theta_0}(\boldsymbol{x})\|} \cdot \frac{\|\nabla_\theta f_{\theta_0}(\boldsymbol{x}')\|}{\|\nabla_\theta \ell(\boldsymbol{x}', y'; f_{\theta_0})\|} \cdot \nabla_\theta \ell(\boldsymbol{x}', y'; f_{\theta_0}) \right\|.
\end{aligned}
\tag{19}
$$

Rearranging some terms, we get a gradient distance term $\left\| \nabla_\theta \ell(\boldsymbol{x}, y; f_{\theta_0}) - \nabla_\theta \ell(\boldsymbol{x}', y'; f_{\theta_0}) \right\|$ in a lower bound of the FORGE embedding distance:

$$
\begin{aligned}
&\|\phi(\boldsymbol{x}, y, f_{\theta_0}, \ell) - \phi(\boldsymbol{x}', y', f_{\theta_0}, \ell)\| \\
&\geq \frac{\|\nabla_\theta f_{\theta_0}(\boldsymbol{x})\|}{\|\nabla_\theta \ell(\boldsymbol{x}, y; f_{\theta_0})\|} \cdot \left\| \nabla_\theta \ell(\boldsymbol{x}, y; f_{\theta_0}) - \nabla_\theta \ell(\boldsymbol{x}', y'; f_{\theta_0}) \right\| \\
&\quad - \frac{\|\nabla_\theta f_{\theta_0}(\boldsymbol{x})\|}{\|\nabla_\theta \ell(\boldsymbol{x}, y; f_{\theta_0})\|} \left\| \nabla_\theta \ell(\boldsymbol{x}', y'; f_{\theta_0}) - \frac{\|\nabla_\theta \ell(\boldsymbol{x}, y; f_{\theta_0})\|}{\|\nabla_\theta f_{\theta_0}(\boldsymbol{x})\|} \cdot \frac{\|\nabla_\theta f_{\theta_0}(\boldsymbol{x}')\|}{\|\nabla_\theta \ell(\boldsymbol{x}', y'; f_{\theta_0})\|} \cdot \nabla_\theta \ell(\boldsymbol{x}', y'; f_{\theta_0}) \right\|.
\end{aligned}
\tag{20}
$$

With Equation 20, if we achieve $\epsilon$-cover over FORGE embeddings, $\|\phi(\boldsymbol{x}, y, f_{\theta_0}, \ell) - \phi(\boldsymbol{x}', y', f_{\theta_0}, \ell)\| \leq \epsilon$, the gradient embedding also has a bounded coverage:

$$
\left\| \nabla_\theta \ell(\boldsymbol{x}, y; f_{\theta_0}) - \nabla_\theta \ell(\boldsymbol{x}', y'; f_{\theta_0}) \right\| \leq \diamond \cdot \epsilon + \bigcirc
\tag{21}
$$

where $\diamond = \max_{\boldsymbol{x}, y \in \mathcal{S}_{\text{FORGE}}} \frac{\|\nabla_\theta \ell(\boldsymbol{x}, y; f_{\theta_0})\|}{\|\nabla_\theta f_{\theta_0}(\boldsymbol{x})\|}$ and $\bigcirc = \max_{\boldsymbol{x}', y', \boldsymbol{x}, y \in \hat{\mathcal{S}}_{FORGE}} \left\| \nabla_\theta \ell(\boldsymbol{x}', y'; f_{\theta_0}) - \frac{\|\nabla_\theta \ell(\boldsymbol{x}, y; f_{\theta_0})\|}{\|\nabla_\theta f_{\theta_0}(\boldsymbol{x})\|} \cdot \frac{\|\nabla_\theta f_{\theta_0}(\boldsymbol{x}')\|}{\|\nabla_\theta \ell(\boldsymbol{x}', y'; f_{\theta_0})\|} \cdot \nabla_\theta \ell(\boldsymbol{x}', y'; f_{\theta_0}) \right\|$.

Recalling the upper bound of the loss reduction gap, if we use the BADGE embedding, the upper bound $\Gamma_{\text{BADGE}}$ is

$$
\begin{aligned}
&r_{0 \to T}(\boldsymbol{x}, y; f_\theta^{\text{lin}}) - r_{0 \to T}(\boldsymbol{x}', y'; f_\theta^{\text{lin}}) \\
&\leq \epsilon \|\Delta\theta_T\| + 2 \max_{\boldsymbol{x}', \boldsymbol{x} \in \hat{\mathcal{S}}_{\text{BADGE}}} (\|\nabla_\theta f_{\theta_0}(\boldsymbol{x})\|, \|\nabla_\theta f_{\theta_0}(\boldsymbol{x}')\|)^2 \|\Delta\theta_T\|^2.
\end{aligned}
\tag{22}
$$

With FORGE embedding, the upper bound $\Gamma_{\text{FORGE}}$ is

$$
\begin{aligned}
&r_{0 \to T}(\boldsymbol{x}, y; f_\theta^{\text{lin}}) - r_{0 \to T}(\boldsymbol{x}', y'; f_\theta^{\text{lin}}) \\
&\leq (\diamond \cdot \epsilon + \bigcirc) \|\Delta\theta_T\| + 2 \max_{\boldsymbol{x}', \boldsymbol{x} \in \hat{\mathcal{S}}_{\text{FORGE}}} (\|\nabla_\theta f_{\theta_0}(\boldsymbol{x})\|, \|\nabla_\theta f_{\theta_0}(\boldsymbol{x}')\|)^2 \|\Delta\theta_T\|^2.
\end{aligned}
\tag{23}
$$

If $\gamma_{\text{BADGE}} > \gamma_{\text{FORGE}}$, the upper bound $\Gamma_{\text{FORGE}}$ with FORGE embedding is smaller when

$$
\begin{aligned}
\epsilon + 2\gamma_{\text{BADGE}} \|\Delta\theta_T\| &> \diamond \cdot \epsilon + \bigcirc + 2\gamma_{\text{FORGE}} \|\Delta\theta_T\| \\
\|\Delta\theta_T\| &> \frac{(\diamond - 1) \cdot \epsilon + \bigcirc}{2(\gamma_{\text{BADGE}} - \gamma_{\text{FORGE}})}.
\end{aligned}
\tag{24}
$$

$\square$

# C Experiments

## C.1 Baselines

**Random** Randomly selecting a subset of samples without repetition.

**Margin**  Selecting the data samples that are closest to their decision margin and have high uncertainty. The distance to the decision margin is measured by the difference between the largest logit and the second-largest logit.

**DUC**  (Xie et al., 2023) The authors first utilize Dirichlet-based uncertainty calibration (DUC) to mitigate mis-calibration of neural networks under distribution shifts. Then, they use a two-round procedure to select data samples with high distribution uncertainty and high data uncertainty. Distribution uncertainty helps identify data samples that are out of the source domain, and data uncertainty captures discriminative samples.

**CORESET**  (Sener & Savarese, 2018) Selecting diverse data samples using a k-center greedy algorithm in an embedding space. They use the last hidden state to construct embeddings.

**BADGE**  (Ash et al., 2020) Selecting diverse data samples using a k-means++ algorithm in an embedding space. They use the last layer's gradient to construct embeddings.

**CLUE**  (Prabhu et al., 2021) They use the last hidden state to construct embeddings and then run k-means clustering in an embedding space. The uncertainty, measured in predictive entropy, serves as the weight of the k-means clustering. The data samples that are closest to each clustering center are selected.

### C.2  FORGE Embedding Computation

In practice, we compute the feature norm (RHS of Equation 4, which serves as a principled upper bound on prediction variability for a fixed parameter update. The feature norm is particularly suitable for data selection because the actual parameter update is not yet known at that stage. As detailed in Section 5.2, we estimate it by extracting last-layer embeddings from neural networks across tasks, and use these to construct the FORGE embedding.

### C.3  Dataset Setup

Our method assumes a pool-based scenario, where the full set of online data is available prior to selection for labeling. This setting is common in many real-world applications where data can be collected in batches before annotation begins, such as autonomous driving and document classification where datasets are periodically updated. We split the data evenly across domains during training, which consistently stabilizes training. For example, in VLCS (4 domains), when the source domain is Caltech and the total batch size is 16, we allocate 4 samples to the source domain in each batch.

### C.4  Computational Efficiency

We compared our forward-only approach for constructing FORGE embeddings using Resnet-50 and the VLCS dataset against a baseline requiring a backward pass (batch size = 16, 100 batches). Our method reduces embedding construction time from 37.1 seconds to 12.7 seconds — a 2.9 $\times$ speedup.

## D  Limitation

Although this study illustrates the potential benefits of incorporating prediction variability into active adaptation problems, several important limitations remain unaddressed. In particular, outliers and class imbalance frequently occur in live traffic scenarios, yet the current formulation does not explicitly account for these factors. A thorough consideration of such challenges—along with corresponding mitigation strategies—would substantially enhance the robustness and practical applicability of the proposed method.

