# OpenReview forum: "Improving Single-round Active Adaptation: A Prediction Variability Perspective"
_TMLR — Accepted by TMLR_

### Review · Reviewer_JXb8 · 2025-06-26

**Summary Of Contributions:**

This paper addresses the problem of single-round active adaptation, where a model must select a subset of online data in a single step and use it for continuous training to adapt to distribution shifts, thereby minimizing annotation turnaround time.
The authors mention that traditional data selection methods focusing solely on diversity are insufficient for this setting, and introduce the concept of prediction variability—the degree to which a sample’s prediction changes during training—as a critical factor.
They develop a theoretical framework centered on the loss reduction gap and show that minimizing this gap requires considering both gradient distance and prediction variability.
To address this, they propose FORGE (Feature-norm scaled Gradient Embedding), a method that rescales gradient embeddings by each sample's variability, enabling diversity-based selection algorithms to better prioritize impactful data.
FORGE is computationally efficient, requiring only forward passes, and is applicable across various tasks, including image classification, sentiment analysis, question answering, and reward modeling.
Extensive experiments demonstrate that FORGE consistently outperforms both traditional baselines and, surprisingly, even random selection, highlighting the importance of accounting for prediction variability in single-round adaptation.

**Audience:**

Yes

**Claims And Evidence:**

Yes

**Requested Changes:**

It would be helpful if the authors could elaborate further on how the prediction variability is calculated in practice. While the paper provides a theoretical formulation using tangent feature norms, a more concrete explanation or step-by-step procedure—especially in the context of their analytical approximation—would improve clarity and reproducibility. Additionally, the paper should clarify the data access assumptions in the single-round setting: does the algorithm have access to the full pool of online data before selecting a subset to be labeled, or is the data arriving in a streaming fashion? If the latter, estimating prediction variability on unseen future examples could pose significant challenges. Addressing how FORGE could operate under streaming constraints—or explicitly stating that it assumes a pool-based scenario—would strengthen the applicability of the method.

**Strengths And Weaknesses:**

strength: This paper addresses an important problem in active learning, how to minimize the time human annotators need to spend (aka single round active adaptation), which is particularly relevant in large-scale systems like large language models, where repeated querying and annotation are costly due to high inference latency and data volume. The authors make a compelling case for reducing annotation turnaround time without compromising adaptation quality. A major strength of the work lies in its theoretical depth: the paper introduces prediction variability as a critical yet previously underappreciated factor in data selection for adaptation, complementing traditional diversity-based approaches. This idea is both novel and well-supported by analytical insights and empirical evidence. Furthermore, the proposed method, FORGE, is designed with computational efficiency in mind and is shown to be robust across a wide range of tasks, including image classification, sentiment analysis, and reward modeling.

Weakness: The paper is rather technical and lacks accessible, real-world examples that could help readers. Additionally, the work would benefit from a deeper analysis of the trade-offs between computational cost and performance improvement, particularly in terms of annotation budget and model training time. Another limitation is the minimal ablation or sensitivity analysis of key components such as the impact of feature norm scaling or alternative embedding formulations. Without this, it is difficult to fully assess the robustness and generalizability of the approach across different settings.

---

> ### Author Response · Authors · 2025-08-13
> **Author Response**
>
> We thank the reviewer for the helpful suggestions.
>
> **The paper is rather technical and lacks accessible, real-world examples that could help readers.**
>
> We appreciate the concern regarding the technical depth of our paper and the need for more accessible, real-world examples. To clarify, we motivate our work with several concrete scenarios:
>
> * A vision model in a surveillance camera or autonomous driving system encountering new, unseen images daily;
>
> * A fraud detection model processing transactions from new users or adversaries attempting novel attacks;
>
> * A language model in a chat application receiving time-sensitive questions that evolve over time.
>
> These examples illustrate how distribution shifts naturally arise in diverse, mission-critical applications and lead to significant performance degradation—such as traffic accidents, financial loss, or incorrect medical advice—highlighting the importance of effective adaptation strategies.
>
> While these motivating cases are included, we acknowledge that emphasizing them more prominently could improve the accessibility of our presentation. We will revise the manuscript to bring these examples to the forefront, helping readers better connect the technical content to real-world impact.
>
> **Additionally, the work would benefit from a deeper analysis of the trade-offs between computational cost and performance improvement, particularly in terms of annotation budget and model training time.**
>
> In our empirical study, increased computational resources—both in terms of annotation budget and model training time—consistently lead to improved performance. Therefore, rather than a trade-off, our study highlights how leveraging more annotation and training efforts can effectively boost model adaptation.
>
> **Another limitation is the minimal ablation or sensitivity analysis of key components such as the impact of feature norm scaling or alternative embedding formulations. Without this, it is difficult to fully assess the robustness and generalizability of the approach across different settings.**
>
> We performed an ablation on VLCS dataset for the norm scaling factor in our FORGE embedding construction, considering the ½, 1, and 2 powers of the norm term:
>
> | Method   | 5%       | 10%      |
> |----------|----------|----------|
> | norm^½   | 74.63    | 75.16    |
> | **norm^1**   | **75.19**| **76.06**|
> | norm^2   | 74.25    | 75.83    |
>
> **It would be helpful if the authors could elaborate further on how the prediction variability is calculated in practice.**
>
> In practice, we compute the feature norm $||\nabla_{\theta} f_{\theta_0}(\mathbf {x})||$ (RHS of Equation (4)), which serves as a principled upper bound on prediction variability for a fixed parameter update. The feature norm is particularly suitable for data selection because the actual parameter update is not yet known at that stage. As detailed in Section 5.2, we estimate it by extracting last-layer embeddings from neural networks across tasks, and use these to construct the FORGE embedding. This approach has proven effective in our experiments, and we would be glad to further illustrate the procedure if helpful.
>
> **Additionally, the paper should clarify the data access assumptions in the single-round setting.**
>
> Our method assumes a pool-based scenario, where the full set of online data is available prior to selection for labeling. This setting is common in many real-world applications where data can be collected in batches before annotation begins, such as autonomous driving and document classification where datasets are periodically updated.

---

> > ### Author Response · Authors · 2025-09-13
> > **Author Comment**
> >
> > Dear Reviewer JXb8,
> >
> > If you have any remaining concerns, please let us know.
> >
> > Authors

---

### Review · Reviewer_tV17 · 2025-06-30

**Summary Of Contributions:**

The paper proposes a method for active adaptation (fine-tuning) of pre-trained models based on a single-round selection approach. The method is based on calculating an embedding for each sample and then performing selection using the k-center greedy approach. The embedding is constructed by rescaling the gradient embedding using a feature norm. This choice of embedding is based on theoretical derivation that shows that such embedding allows for selecting samples with high prediction variability, which is crucial for good selection. The proposed method is evaluated on four different tasks and seven datasets, where it consistently shows improvements over the random selection and outperforms existing active adaptation approaches.

**Audience:**

Yes

**Broader Impact Concerns:**

The work does not have any ethical concerns that would need to be addressed in a broader impact statement.

**Claims And Evidence:**

Yes

**Requested Changes:**

*Critical*
- The paper should extend the evaluation to include additional active learning methods and include random and CLUE into all tables.

*To strengthen the work*
- Improving the presentation of the paper. E.g., making the use of $n$ or $N$ consistent throughout the paper, clarifying the meaning of things in plots, etc.

**Strengths And Weaknesses:**

**Strengths**
- The theoretical analysis is sound and clearly motivates the proposed approach.
	- The theoretical analysis is based on estimating the upper bound of the loss reduction gap.
	- The paper shows that the loss reduction gap is upper bounded by the sum of gradient distance and the variability upper bound.
	- The variability upper bound implies that without considering it, an active learning method may neglect the samples that are difficult to learn by learning its close neighbors.
	- This variability upper bound term is then used to construct the proposed embedding used for selection. The embedding contains a feature norm factor that captures the variability upper bound under unknown training dynamics.
- The evaluation is extensive and shows that the proposed approach works well for a wide range of tasks.
	- The evaluation is performed on four tasks and seven datasets where the proposed method outperforms other approaches.
- The paper is clearly written and easy to follow.

**Weaknesses**
- The evaluation could be improved.
	- Why does Table 1 not show the results of Random or other methods, such as CLUE?
	- Why does Table 4 compare only with Random and BADGE and not with other methods such as CLUE, CORESET, etc?
	- The paper should include a comparison with more models, e.g., TypiClust [1], ProbCover [2], and BiLAF [3].
- The presentation of the paper could be improved.
	- In Algorithm 1 line 4 there should be $i$ in argmax, i.e., $i \in [N] \setminus \mathbf{s}$.
	- The paper flips between using $n$ and $N$ for the number of samples. E.g., Algorithm 1 uses $N$ while Proposition 2 uses $n$. This should be made consistent throughout the paper.
	- It is unclear what the meaning of the red line in Figure 5 is.
	- In the proof of Proposition 2 $c_i$ should be $c_j$ and in the last line of the proof $1/n$ should be $1/N$. Additionally, using index $j$ both for looping over sums and to mean the closest selected neighbor makes it harder to read the proof.
	- In the proof of Theorem 4 $l(\mathbf{x},y;f^{\text{lin}}_{\theta_T}) - l(\mathbf{x},y;f_{\theta_0})$ should be $l(\mathbf{x},y;f_{\theta_0}) - l(\mathbf{x},y;f^{\text{lin}}_{\theta_T})$.
- Typos:
	- Section 5.2. last sentence, "neural network and is, therefore. computational efficient." should be "neural network and is, therefore, computationally efficient."
	- Section 6.1. "Model architecture. we use the ..." should be "Model architecture. We use the ..."

*References*

[1] Guy Hacohen, Avihu Dekel, Daphna Weinshall, "Active Learning on a Budget: Opposite Strategies Suit High and Low Budgets", In ICML, 2022

[2] Ofer Yehuda, Avihu Dekel, Guy Hacohen, Daphna Weinshall, "Active Learning Through a Covering Lens", In NeurIPS, 2022

[3] Han Lu, Yichen Xie, Xiaokang Yang, Junchi Yan, "Boundary Matters: A Bi-Level Active Finetuning Framework", In NeurIPS, 2024

---

> ### Author Response · Authors · 2025-08-13
> **Author Response**
>
> We thank the reviewer for the helpful suggestions.
>
> **Why does Tables 1 and 4 not show the results of Random or other methods, such as CLUE?**
>
> We have updated Table 1 to include both Random and CLUE. For Table 4, since CLUE relies on prediction entropy (unavailable in reward modeling tasks), we include Coreset instead, ensuring a fair and meaningful comparison.
>
> **Table 1**
>
> | Method | Selected Train | Unselected Train | Validation | Test |
> |--------|----------------|------------------|------------|------|
> | Random | 100 | 74.30 | 74.52 | 73.89 |
> | Coreset | 100 | 73.68 | 73.67 | 72.98 |
> | Badge | 100 | 75.18 | 74.15 | 73.28 |
> | CLUE | 100 | 75.90 | 73.77 | 72.97 |
> | **Forge (ours)** | 100 | **76.13** | **75.19** | **74.94** |
>
> **Table 4**
>
> | Method | Accuracy |
> |--------|-------|
> | Random | 63.01 |
> | Coreset | 63.89 |
> | BADGE | 63.12 |
> | **FORGE (ours)** | **64.45** |
>
> **The paper should include a comparison with more models, e.g., TypiClust [1], ProbCover [2], and BiLAF [3].**
>
> We conducted additional experiments, which further confirm that our method consistently outperforms competitors in the proposed single-round active adaptation setting.
>
> | Method | Image-CLS on VLCS 5% | Image-CLS on VLCS 10% | Image-CLS on VisDA 5% | Image-CLS on VisDA 10% | Sentiment-CLS 5% | Sentiment-CLS 10% | Span-QA 5% | Span-QA 10% |
> |--------|----------------------|-----------------------|-----------------------|------------------------|------------------|-------------------|------------|-------------|
> | **FORGE (ours)** | **75.19** | **76.06** | **82.28** | **85.45** | **51.79** | **52.23** | **38.67** | **39.06** |
> | TypiClust | 74.94 | 75.23 | 81.75 | 85.05 | 51.38 | 51.94 | 38.38 | 38.87 |
> | ProbCover | 73.26 | 73.69 | 81.55 | 84.88 | 50.90 | 51.58 | 38.19 | 38.41 |
> | BiLAF | 74.87 | 75.02 | 81.38 | 84.44 | 50.47 | 50.78 | 37.94 | 38.12 |
>
>
> **The presentation of the paper could be improved.**
>
> Thank you for the helpful suggestions. We will revise the paper to further enhance its presentation and clarity. In addition, we will remove the red line in Figure 5 to prevent potential confusion and instead convey the key observations clearly through the caption.

---

> > ### Comment · Reviewer_tV17 · 2025-09-03
> > **Rebuttal answer**
> >
> > Dear Authors,
> >
> > Your rebuttal has addressed the major concerns I had in the initial review. Furthermore, after reading other reviews and answers, I find that the rebuttal has successfully addressed the concerns raised.
> >
> > However, the manuscript was not updated with the changes proposed by the reviewers. Before proposing my final recommendation, I would like to see the updated version of the manuscript.
> >
> > Reviewer tV17

---

> > > ### Author Response · Authors · 2025-09-11
> > > **Author Response**
> > >
> > > Dear reviewer tV17,
> > >
> > > We have updated the manuscript to incorporate the requested changes from the rebuttal phase. Specifically, we now use $N$ to denote the number of samples, use $i$ exclusively for looping over sums to avoid confusion, and introduce $x_{s,i}$ to represent the closest selected neighbor, thereby improving the clarity and readability of the proof of Proposition 2.
> > >
> > > Authors

---

> > > > ### Comment · Reviewer_tV17 · 2025-09-11
> > > > **Revision response**
> > > >
> > > > Dear Authors,
> > > >
> > > > Thank you for providing the revision. The revision successfully incorporates all the changes requested by the reviewers.
> > > >
> > > > As a minor comment for future submissions, it would be better if these changes were colored, making it easier for reviewers to check them.
> > > >
> > > > Reviewer tV17

---

> > > > > ### Author Response · Authors · 2025-09-12
> > > > > **Author Response**
> > > > >
> > > > > Dear Reviewer tV17,
> > > > >
> > > > > We genuinely appreciate your suggestions for improving the quality of our submission. We will keep them in mind for future work.
> > > > >
> > > > > Authors

---

### Review · Reviewer_qHgt · 2025-07-30

**Summary Of Contributions:**

This paper introduces a novel active learning framework tailored for the **single-round active adaptation** setting, where a model selects a subset of unlabeled data once and then trains continuously. Unlike traditional multi-round strategies, this formulation is better suited for online or dynamic learning scenarios. The paper proposes a principled approach for data selection that jointly considers **gradient distance** (important in early training) and **prediction variability** (which becomes dominant in later stages). To capture both aspects, it develops a new embedding technique called **FORGE (Feature-norm Scaled Gradient Embedding)**, which scales gradient embedding by the prediction variability of samples. This promotes the selection of **high-variability samples**—often underrepresented in prior diversity-based methods. The paper presents theoretical analyses showing that reducing the upper bound of  **loss reduction gap** between selected and unselected samples leads to improved expected loss reduction. The **upper bound loss reduction gap** is estimated by encompassing a gradient distance term and prediction variability term.  It also proves that FORGE achieves a **tighter lower bound** on this quantity compared to existing embedding techniques such as BADGE. It further provides an efficient implementation of FORGE using only single forward pass, making it computationally practical across diverse tasks. The proposed method is evaluated across multiple domains—including vision, sentiment classification, span-based QA, and reward modeling—demonstrating consistent improvements over strong baselines such as CORESET, BADGE, and uncertainty-based strategies.

**Audience:**

Yes

**Claims And Evidence:**

Yes

**Requested Changes:**

- In page number 11, first paragraph, second line, the authors use only selected samples to calculate gradient distance for BADGE. Maybe it is a notation error. It should account for both selected and unselected samples.

- There are also some typos or misspellings.

- Ablation studies need to be done.

- Major shortcomings of the proposed approach should be discussed.

**Strengths And Weaknesses:**

-  The paper introduces an active adaptation algorithm framework by performing diversity based data selection using k-center greedy algorithm. Each data is represented by a novel embedding function called FORGE(Feature-norm Scaled Gradient Embedding). FORGE embedding achieves low prediction variability and a small gradient distance simultaneously.

- It considers both high and low-variability data. Does it take into account the exception cases like outliers?

- The paper evaluates the performance of FORGE embeddings by using diversity based data selection only. However, its results utilizing uncertainty based selection remains unknown.

-  It performs the analysis on a single round active adaptation problem by formulating the expected loss reduction objectives as minimizing the upper bound of a maximum loss reduction gap. The upper bound loss reduction gap is estimated by encompassing a gradient distance term and prediction variability term. Further analysis also shows that a given approach achieves a higher lower bound of the expected loss reduction.
- The study mentions an interesting observation regarding higher-order variability terms relying on first order tangent features instead of higher-order gradients. Are there any experiment analyses to support this fact?

- It also provides analytical constructions of FORGE embedding using only a single forward pass of a neural network. It incorporates last layer gradient embedding instead of computing initial tangent feature and gradient embedding to alleviate the computational overhead. However, It doesn’t provide any quantitative analysis to support the fact.

- Extensive results on various vision and language tasks verify the effectiveness of the proposed approach for active learning. However, the experiment relies on source data to finetune the model first and later adapt it into target domain using 5 to 10% data selected using active learning. It is not discussed how much source data is enough to stabilize training and improved adaptation.

- This paper lacks additional ablation studies to fully support the experiment. Furthermore, it only takes accuracy as the evaluation measures to compare the performance despite having several metrics for assessment. Additionally, how the proposed active learning strategy mitigates the effects of class imbalance during query selections should also be addressed?

---

> ### Author Response · Authors · 2025-08-13
> **Author Response**
>
> We thank the reviewer for the helpful suggestions.
>
> **It considers both high and low-variability data. Does it take into account the exception cases like outliers?**
>
> While our primary focus is modeling prediction variability, our approach is compatible with strategies for handling outliers. In particular, the Core-Set paper’s mixed-integer programming (MIP) formulation of the k-center problem improves robustness to outliers, and—since both works build on the k-center algorithm—can be directly incorporated into our framework if needed [1].
>
> **However, its results utilizing uncertainty based selection remains unknown.**
>
> Thank you for this important point. Our work specifically investigates diversity-based methods, which we show yield stronger empirical results than uncertainty-based alternatives in our empirical analysis. However, we agree with the reviewer that uncertainty-based approaches are worth further exploration.
>
> **The study mentions an interesting observation regarding higher-order variability terms relying on first order tangent features instead of higher-order gradients. Are there any experiment analyses to support this fact?**
>
> This observation is supported by prior work on the neural tangent kernel (NTK) regime [2, 3], a well-established framework in deep learning theory [4]. These studies demonstrate that, under certain conditions, neural networks evolve like linear models during training, allowing higher-order variability terms to be characterized using first-order tangent features.
>
> **It incorporates last layer gradient embedding instead of computing initial tangent feature and gradient embedding to alleviate the computational overhead. However, It doesn’t provide any quantitative analysis to support the fact.**
>
> We compared our forward-only approach against a baseline requiring a backward pass (batch size = 16, 100 batches). Our method reduces embedding construction time from 37.1 s to 12.7 s—a 2.9x speedup.
>
> **It is not discussed how much source data is enough to stabilize training and improved adaptation.**
>
> We split the data evenly across sources during training, which consistently stabilizes training. For example, in VLCS (4 domains), when the source domain is Caltech and the total batch size is 16, we allocate 4 samples to the source domain in each batch.
>
> **Furthermore, it only takes accuracy as the evaluation measures to compare the performance despite having several metrics for assessment.**
>
> For the QA benchmark, we additionally report the F1 score. Our model remains the top performer under this metric:
>
> | Method         | 5%     | 10%     |
> |----------------|--------|---------|
> | Random         | 54.03  | 54.73   |
> | Coreset        | 53.08  | 54.15   |
> | BADGE          | 54.12  | 54.70   |
> | FORGE (ours)   | **54.20** | **54.79** |
>
> **Ablation studies need to be done.**
>
> We performed an ablation on VLCS dataset for the norm scaling factor in our FORGE embedding construction, considering the ½, 1, and 2 powers of the norm term:
>
> | Method   | 5%       | 10%      |
> |----------|----------|----------|
> | norm^½   | 74.63    | 75.16    |
> | norm^1   | **75.19**| **76.06**|
> | norm^2   | 74.25    | 75.83    |
>
> **Additionally, how the proposed active learning strategy mitigates the effects of class imbalance during query selections should also be addressed?**
>
> Our method targets prediction variability rather than class imbalance. We refer to related work [5, 6] that specifically addresses active learning under class-imbalanced settings.
>
> **In page number 11, first paragraph, second line, the authors use only selected samples to calculate gradient distance for BADGE. There are also some typos or misspellings.**
>
> Thanks for pointing out the typo. We will correct the typo on page 11: one instance of $x$ should be $x'$, representing an unselected sample. In addition, we will revise the manuscript to address grammar issues.
>
> **Major shortcomings of the proposed approach should be discussed.**
>
> Although the study illustrates the potential benefits of incorporating prediction variability into active adaptation problems, several important limitations remain unaddressed. In particular, outliers and class imbalance frequently occur in live traffic scenarios, yet the current formulation does not explicitly account for these factors. A thorough consideration of such challenges—along with corresponding mitigation strategies—would substantially enhance the robustness and practical applicability of the proposed method.
>
> **Reference**
>
> [1] Active Learning for Convolutional Neural Networks: A Core-Set Approach. ICLR 2018.
>
> [2] Neural tangent kernel: convergence and generalization in neural networks. ICML 2018.
>
> [3] Wide neural networks of any depth evolve as linear models under gradient descent. JSM 2019.
>
> [4] A kernel-based view of language model fine-tuning. ICML 2023.
>
> [5] Active Learning for Imbalanced Datasets. WACV 2020.
>
> [6] Active learning for class imbalance problem. SIGIR 2007.

---

> > ### Comment · Reviewer_qHgt · 2025-09-04
> >
> > Thank you for your response. Most of my concerns are addressed.

---

### Decision · Action_Editor_NS9q · 2025-09-24

**Recommendation:** Accept as is

**Audience:**

Yes

**Audience Explanation:**

This paper addresses the topic of Single Round Active Adaptation of Trained Models, which is an important research problem and is of interest to several members in TMLR's audience.

**Claims And Evidence:**

Yes

**Claims Explanation:**

In this work, the authors proposed a framework to actively adapt pre-trained models based on single-round selection. One of the main contributions is to theoretically show that analysing prediction variability of each data sample throughout the training is important and can be effectively used for this task, along with the widely used data diversity.

Detailed experimental results and comparisons with the existing approaches in literature is reported. The method is theoretically founded and empirically preforms better or at par with existing techniques, thereby making it interesting to the research community. One minor issue is that the improvements over the  state-of-the-art method is not significant. But overall, the claims are supported and convincing.